# Apoptotic tumor cell-derived microRNA-375 uses CD36 to alter the tumor-associated macrophage phenotype

Ann-Christin Frank [1], Stefanie Ebersberger[2], Annika F. Fink[1], Sebastian Lampe[1], Andreas Weigert [1], Tobias Schmid[1], Ingo Ebersberger[3,4], Shahzad Nawaz Syed[1] & Bernhard Brüne [1,5]

Tumor-immune cell interactions shape the immune cell phenotype, with microRNAs (miRs) being crucial components of this crosstalk. How they are transferred and how they affect their target landscape, especially in tumor-associated macrophages (TAMs), is largely unknown. Here we report that breast cancer cells have a high constitutive expression of miR-375, which is released as a non-exosome entity during apoptosis. Deep sequencing of the miRome pointed to enhanced accumulation of miR-375 in TAMs, facilitated by the uptake of tumor-derived miR-375 via CD36. In macrophages, miR-375 directly targets *TNS3* and *PXN* to enhance macrophage migration and infiltration into tumor spheroids and in tumors of a xenograft mouse model. In tumor cells, miR-375 regulates CCL2 expression to increase recruitment of macrophages. Our study provides evidence for miR transfer from tumor cells to TAMs and identifies miR-375 as a crucial regulator of phagocyte infiltration and the subsequent development of a tumor-promoting microenvironment.

[1] Faculty of Medicine, Institute of Biochemistry I, Goethe-University Frankfurt, Theodor-Stern-Kai 7, 60590 Frankfurt, Germany. [2] Institute of Molecular Biology gGmbH, Ackermannweg 4, 55128 Mainz, Germany. [3] Department for Applied Bioinformatics, Institute for Cell Biology and Neuroscience, Goethe-University Frankfurt, Max-von-Laue Strasse 13, 60438 Frankfurt, Germany. [4] Senckenberg Biodiversity and Climate Research Centre Frankfurt (BIK-F), Frankfurt 60325, Germany. [5] German Cancer Research Consortium (DKTK), Partner Site, Frankfurt 60590, Germany. These authors jointly supervised this work: Shahzad Nawaz Syed and Bernhard Brüne. Correspondence and requests for materials should be addressed to S.N.S. (email: syed@biochem.uni-frankfurt.de) or to B.B. (email: b.bruene@biochem.uni-frankfurt.de)

The breast cancer microenvironment consists of not only tumor cells but also of stromal cells, including distinct immune cell subsets. Among them, tumor-associated macrophages (TAMs) stand out both in their tumor-promoting ability and in their prevalence as well[1,2]. Due to their high plasticity, macrophages (MΦ) can undergo coordinated changes in gene expression in response to tumor microenvironmental cues such as apoptotic cells, which polarizes them toward a pro-tumoral phenotype with anti-inflammatory and immunosuppressive properties[3,4]. These pro-tumoral MΦ not only support tumor survival and growth but also contribute to metastasis, tumor angiogenesis, and immune evasion[5]. In patients with solid tumors, such as prostate, ovarian, cervical, and breast cancer, a high number of infiltrating TAMs correlated with a poor survival prognosis[6]. In breast cancer, TAMs constitute up to 50% of the tumor mass, most of them originating from blood-derived monocytes[1,7]. It is not completely understood how the tumor microenvironment achieves this massive influx of monocytes/MΦ and how it initiates a dramatic and discordant gene expression in TAMs. Understanding this process would be a prerequisite to design therapeutic interventions.

One way tumor cells and immune cells communicate is via microRNAs (miRs), which are noncoding RNAs that inhibit gene expression at the posttranscriptional level[8]. Several studies identified aberrantly expressed miRs involved in many aspects of cancer progression, such as tumor initiation, drug resistance, and metastasis[9]. They are present at abnormal levels in many human tumors[10]. Furthermore, it has been demonstrated that there is an intercellular transfer of miRs between tumor cells and TAMs[11,12], which is mostly ascribed to the release and uptake of extracellular vesicles. However, interestingly, vesicle-encapsulated miRs represent only a minor portion of circulating miRs[13,14]. Hence, how a large number of miRs are transferred between the two cell types is still unknown.

MiR-375 is expressed in several organs and is significantly downregulated in multiple types of cancer, including hepatocellular carcinoma, esophageal carcinoma, gastric cancer, head and neck cancer, melanoma, and glioma[15–19]. Despite the well-characterized role as a tumor suppressor, miR-375 has been found to be upregulated in prostate and notably in breast cancer[20,21]. MiR-375 is highly expressed in estrogen receptor α (ERα)-positive breast tumors, where it creates a positive feedback loop with ERα[21] to foster tumor cell proliferation[22]. Interestingly, baseline expression of miR-375 is negligible in MΦ among stromal cell populations[23]. Here we show accumulation of miR-375 in TAMs and assign a function to this miR as a regulator of MΦ migration by (a) identifying its target genes in TAMs and (b) describing a previously unknown function in tumor cells as a regulator of CCL2 expression. We also discovered an unknown miR-375 transfer mechanism from apoptotic breast cancer cells to TAMs involving CD36, which might pave the way for identifying new drug targets in breast cancer.

## Results

**Coculture with breast cancer cells increases miR-375 in MΦ.** We used a previously established coculture system of MCF-7 cells and human macrophages (MΦ), which mimics the early interaction of tumor and immune cells, provoking tumor cell death followed by engulfment of cell debris by MΦ[24]. The 48 h coculture initiates a pro-tumor phenotype skewing of MΦ, indicated by downregulation of *IL1B*, *IL12*, *CCL18*, CD80, as well as CD163, and upregulation of *CLEC7A* (dectin-1), CD86, CD206, and HLA-DR (Supplementary Fig. 1a, b). Using the coculture setup, we follow the global miR expression profile in TAMs. A miRseq-based approach identified 226 miRs that were differentially expressed in TAMs compared with control MΦ (Fig. 1a–c, Supplementary Data 1). miRseq data have been deposited in the ArrayExpress database at EMBL-EBI under accession number E-MTAB-6885. To access sequencing fidelity, resolution-phase MΦ were generated by stimulating human MΦ with resolvin D1 to use them as a reference for miRseq of TAMs, which corroborated a previously described induction of miR-146 and miR-219[25] (Supplementary Fig. 2). Among the miRs identified in the MCF-7-MΦ coculture miRseq, miR-375 emerged as most highly elevated compared with control MΦ (Fig. 1b, c), which was validated by quantitative PCR (qPCR) (Fig. 1d). In contrast to the coculture with tumor cells, neither MΦ-polarizing stimuli such as 100 ng/mL LPS (lipopolysaccharide) + 100 U/mL interferon-γ (IFNγ), 20 ng/mL interleukin-4 (IL-4), nor the resolution-phase stimulus resolvin D1 increased miR-375 levels in MΦ (Fig. 1d). We sought to determine the molecular mechanism increasing the miR-375 level in TAMs. First, we checked whether elevated miR-375 in MΦ resulted from de novo transcription triggered by contact with MCF-7 cells. We inhibited de novo synthesis of miR-375 by pretreating MΦ with vehicle or actinomycin D, which blocks transcription. Following 3 h pre-treatment with 2.5 μg/mL actinomycin D, MΦ were cocultured with MCF-7 cells for 24 h and analyzed for miR-375 abundance. Despite transcriptional inhibition, miR-375 levels in MΦ increased upon coculture (Fig. 1e), whereas this was not the case for peroxisome proliferator-activated receptor-γ messenger RNA, which is known to have a short mRNA half-life (Fig. 1f). We further confirmed this finding by knocking down (KD) DICER using small interfering RNA (siRNA) (Fig. 1g), which disrupts the formation of mature miRs in KD cells (Fig. 1h). MCF-7 coculture with DICER KD MΦ increased levels of miR-375 similar to control siRNA coculture MΦ (Fig. 1i). Furthermore, pre-mir-375 expression remained unaltered in coculture MΦ (Fig. 1j), which ruled out de novo expression of miR-375 in MΦ.

It has been well documented that miR-375 is significantly downregulated in multiple types of cancer but highly upregulated in prostate and breast tumors[20–22]. Therefore, we analyzed the abundance of miR-375 in different breast cancer (EFM-192A, MCF-7, T47D, MDA-MB-468, MDA-MB-231, SKBR3, HCC1937), primary human mammary epithelial cells (HMEC; PCS-600-010™), T-cell leukemia (Jurkat), osteosarcoma (143B), malignant melanoma (A375), lung carcinoma (A549), and brain glioblastoma (T98G) cell lines (Supplementary Fig. 3a). There was a high constitutive miR-375 expression in all breast cancer cell lines but not in primary mammary epithelial cells and the other tumor cells tested. To explore specificity of the miR-375 increase in MΦ, we cocultured MΦ with various human breast cancer cell lines that differ in their ER expression (ER + cells: EFM-192A, MCF-7, T47D; ER − cells: MDA-MB-468, MDA-MB-231, SKBR3, HCC1937), HMEC, and Jurkat cells for 24 h (Fig. 1k). MiR-375 abundance was significantly increased in cocultures of MΦ with most of the breast cancer cell lines but not with HCC1937, HMEC, and Jurkat cells, which expressed very low levels of miR-375. As we noticed a correlation between miR-375 content in cancer cells and its increase in MΦ during coculture, we aimed to reduce miR-375 levels in MCF-7 cells. For this purpose, we created a stable miR-375 knockdown (KD; decoy MCF-7 cells) using a miR decoy with an efficacy of around 50% (Supplementary Fig. 3b). In the coculture set-up, these decoy MCF-7 cells provoked only an insignificant miR-375 increase in MΦ (Fig. 1l). Conclusively, the miR-375 increase in MΦ during tumor cell coculture is most probably tumor cell derived.

**MiR-375 is transferred from MCF-7 cells to MΦ.** Considering potential mechanisms that have been proposed for the intercellular transfer of RNA[26], we next explored the pathway for

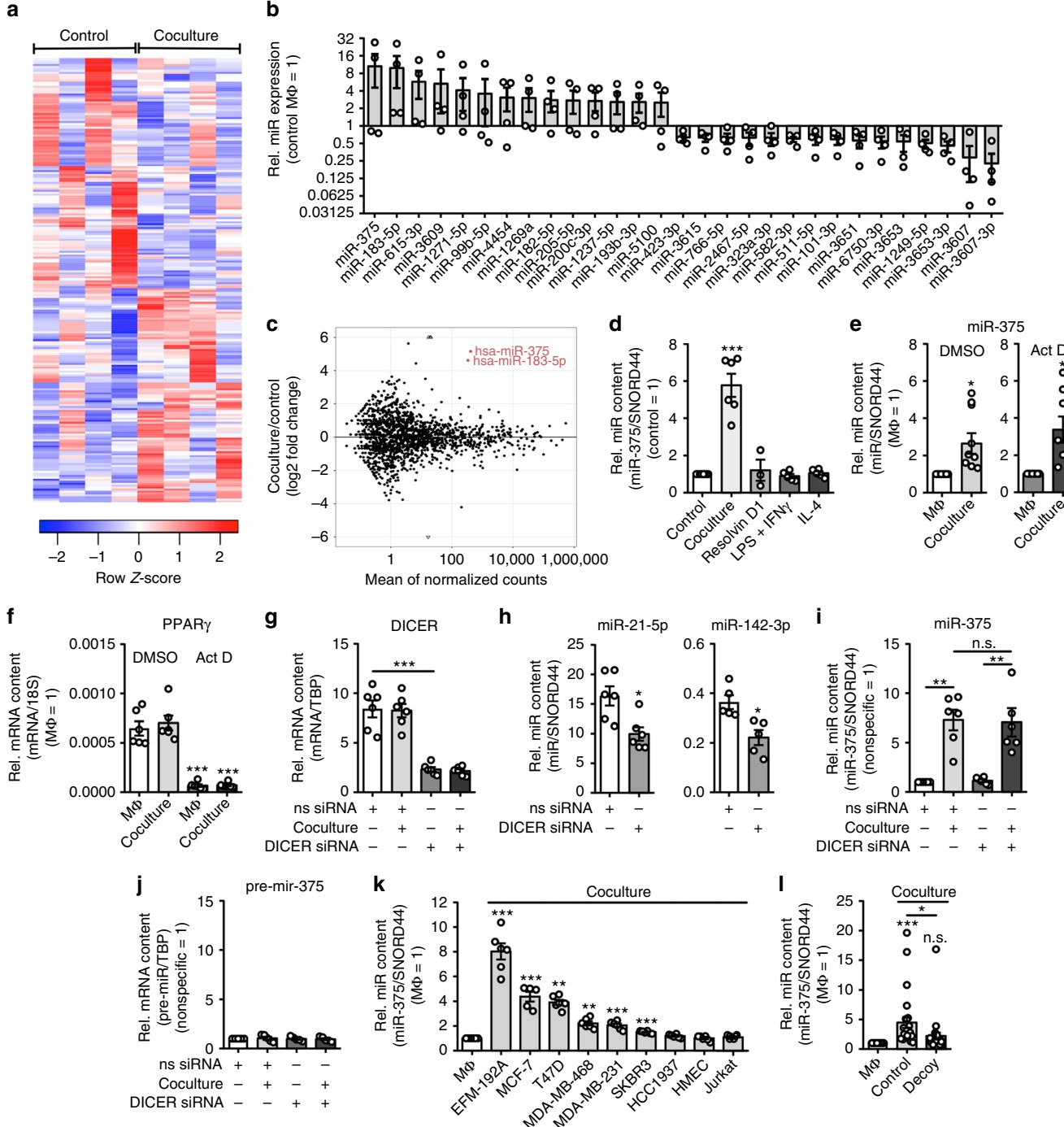

**Fig. 1** Coculture with breast cancer cells increases miR-375 in human MΦ. **a–l** Human PBMC-derived MΦ were used. **a** Heatmap of differentially expressed miRs from control and MΦ cocultured with MCF-7 cells ($n = 4$). **b** Representative differentially expressed miRs in control MΦ vs. cocultured MΦ and **c** MA-plot. **d** MiR-375 abundance was measured in control, 48 h cocultured, polarized (LPS + IFNγ for 24 h, IL-4 for 48 h) and resolution-phase (resolvin D1 for 6 h) MΦ via qPCR, and normalized to untreated MΦ ($n \geq 3$). **e** Primary human MΦ were treated for 3 h with actinomycin D (Act D) or a DMSO control. Cells were washed and cocultured with MCF-7 cells for 24 h. MiR-375 abundance was quantified via qPCR and normalized to MΦ control ($n = 8$). **f** PPARgamma mRNA expression was measured as a positive control ($n \geq 5$). **g–j** MΦ were transfected with nonspecific control (ns siRNA) or DICER siRNA for 24 h and cocultured with MCF-7 cells for another 48 h ($n = 5$–6). **g** DICER mRNA expression in MΦ. **h** Endogenous miR-21-5p and miR-142-3p were measured by qPCR as a control. **i** MiR-375 abundance and **j** pre-miR-375 expression were measured by qPCR in control and cocultured MΦ. **k** MΦ were cocultured with indicated cell lines for 24 h. MiR-375 levels were measured by qPCR and normalized to MΦ control ($n \geq 5$). **l** MΦ were cocultured with MCF-7 control (empty vector transfected) or decoy (miR-375 decoy transfected) cells for 24 h. MiR-375 expression was measured by qPCR and normalized to MΦ control ($n = 27$) using different MCF-7 cell passages. Data of **b** and **d–l** are mean ± SEM and p-values were calculated using two-tailed Student's t-test (**d**, **f**, **g–k**) and one-sample t-test (**e**, **l**). *$p < 0.05$, **$p < 0.01$, ***$p < 0.001$; n.s., not significant

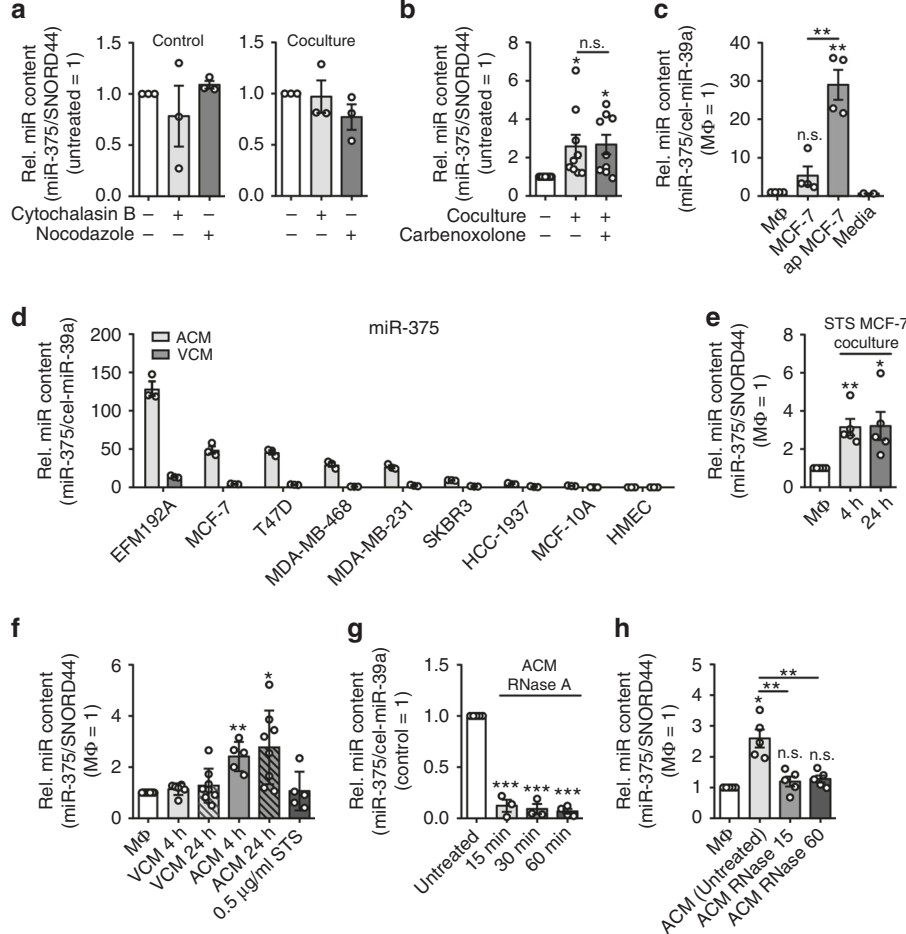

**Fig. 2** MΦ uptake of miR-375 as a non-exosome entity. **a**, **b** Primary human MΦ were cocultured with MCF-7 cells for 24 h. **a** One hour before and during the coculture period, MΦ were treated either with vehicle, cytochalasin B, nocodazole, and **b** carbenoxolone. MiR-375 abundance was quantified via qPCR and normalized to untreated MΦ or untreated coculture MΦ, respectively (n ≥ 3). **c** MiR-375 was measured by qPCR in the supernatants of MΦ, viable MCF-7 cells (MCF-7), STS-treated apoptotic MCF-7 cells (ap MCF-7), media, and normalized to untreated MΦ. Synthetic cel-miR-39a was used as spike-in control (n ≥ 2). **d** VCM and ACM of ER+ (EFM-192A, MCF-7, T47D) and ER− (MDA-MB-468, MDA-MB-231, SKBR3, HCC1937) breast carcinoma cells, mammary epithelial cells (MCF-10A), and primary mammary epithelial cells (HMEC) were analyzed for the abundance of miR-375 (n = 3). **e** MiR-375 level was measured by qPCR in MΦ cocultured with STS-treated apoptotic MCF-7 cells for 4 h. MCF-7 cells were removed from cocultures and MΦ were further cultivated for 20 h (24 h time point). Data are normalized to control MΦ (n = 5). **f** MΦ were treated with STS as a control, or 1:1 diluted supernatants of viable (VCM) or apoptotic (ACM) MCF-7 cells for 30 min. Cells were washed and further cultured in MΦ media for 4 and 24 h, and miR-375 abundance was measured and normalized to untreated MΦ (n ≥ 5). **g** ACM was incubated with control or 50 μg/mL RNase A at 37 °C for indicated time. Before RNA isolation, cel-miR-39a was added as a normalization control. MiR-375 abundance was quantified by qPCR and normalized to control ACM (n ≥ 3). **h** MΦ were incubated for 30 min with either MCF-7 control ACM or RNase A-treated ACM. Cells were washed and cultured for another 24 h in MΦ media. MiR-375 level was determined and normalized to untreated MΦ (n = 5). Data of **a**–**h** are mean ± SEM and p-values were calculated using one-sample t-test. *p < 0.05, **p < 0.01, ***p < 0.001; n.s., not significant

miR-375 uptake into MΦ. One possibility was direct cell-to-cell contacts that require cytoskeletal remodeling[27,28]. Treatment of MΦ with cytochalasin B, which disrupts actin filaments, or nocodazole, which inhibits microtubule polymerization, left miR-375 transfer intact (Fig. 2a). Furthermore, using carbenoxolone to block gap junctions did not alter the miR-375 amount in MΦ after coculture (Fig. 2b). To explore the possibility that miR-375 could be transferred via extracellular vesicles[29–32], we investigated the cell culture supernatants of viable (viable cell conditioned media, VCM) and staurosporine (STS)-treated apoptotic MCF-7 cells (apoptotic conditioned media, ACM) for the abundance of miR-375. Negligible amounts of miR-375 were found in MΦ supernatants or the culture media (Fig. 2c). In contrast, MCF-7 cell VCM, and to a significantly greater extent ACM (~5-fold more as compared to viable MCF-7 and ~30-fold more as compared to MΦ), contained miR-375. Since MCF-7 cells are

caspase-3 deficient, we explored if caspase-3 might alter miR-375 release from apoptotic cells. Therefore, caspase-3-expressing MDA-MB-231 cells were subjected to STS-induced apoptosis and miR-375 release was measured (Fig. 2d), which excludes any influence of caspase-3 in miR-375 release upon apoptosis. Since apoptotic MCF-7 cells release more miR-375, we wondered if we could increase miR-375 in MΦ by culturing them directly with STS-induced apoptotic MCF-7 cells. The coculture of MΦ and STS-treated MCF-7 cells was maintained for 4 h, followed by removal of residual MCF-7 cells to avoid tumor cell necrosis. Already after 4 h, miR-375 was three- to fourfold enhanced in MΦ and remained elevated after 24 h (Fig. 2e). Apparently, the miR-375 increase in MΦ was considerably fast and sustained. Next, we incubated MΦ with VCM and ACM for 30 min and cultured for another 4 h and 24 h in fresh media. Whereas VCM did not alter the miR-375 amount, it was significantly increased

in ACM-treated MΦ after 4 h and 24 h, compared with controls (~3-fold increase) (Fig. 2f). To rule out the possibility that trace amount of STS in the ACM might influence the result, we also treated MΦ with STS alone, which left the miR-375 level in MΦ unaltered (Fig. 2f). Taken together, this points toward a scenario where miR-375 was released in the supernatant of apoptotic tumor cells and was taken up by MΦ by an unknown route.

**Tumor cell-derived miR-375 is taken up by MΦ via CD36**. To understand the transfer of tumor cell-derived miR-375 to MΦ, it was important to determine its physical state, e.g., within exosomes, apoptotic bodies, or lipoprotein bound. Systematic investigation of the physical state of miRs in plasma, serum, and cell culture supernatants has demonstrated that vesicle-encapsulated miRs represent only a small portion of circulating miRs[13,14,33]. Hence, to achieve this objective, ACM was collected and incubated with RNase A for various time points, as exosome-free extracellular miRs have been shown to be sensitive to RNase activity[13] and can bind potential transport proteins of the argonaute-protein family, nucleophosmin, or high- and/or low-density lipoproteins (LDL)[14]. Interestingly, miR-375 detected by qPCR was significantly decreased after RNase A treatment compared with untreated control ACM (roughly 88% reduction after 15 min and 93% reduction after 60 min of treatment) (Fig. 2g). In contrast, miR-183-5p, which was initially found in this study to be similarly elevated in MΦ after coculture with MCF-7 cells (Fig. 1b, c), remained unaffected even after 60 min of RNase A treatment (Supplementary Fig. 4a). This observation was in line with reports, suggesting that miR-183-5p is transferred between cells via extracellular vesicles[34] and thus is protected from RNase A treatment[35]. Importantly, control ACM elevated miR-375 in MΦ, whereas RNase A-treated ACM failed to do so (Fig. 2h). As expected, miR-183-5p was significantly enhanced in MΦ, irrespective of whether ACM was RNase A treated or not (Supplementary Fig. 4b).

To understand transport mechanisms of tumor-derived miR-375, it was incumbent to characterize the miR-375-containing cargo that is taken up by MΦ. Incidentally, apoptotic tumor cells expose oxidized phospholipids (described as oxidized LDL-like sites[36]) on their surface that are recognized by MΦ scavenger receptors such as SRA, LOX1, and CD36[37]. CD36 is widely expressed and may interact with multiple extracellular ligands, including thrombospondin-1, long-chain free fatty acids, oxLDL, advanced glycation end products, and collagens I and IV[38,39]. We prepared ACM of MCF-7 cells in fetal calf serum (FCS), which contains LDL, or in serum-free media. As expected, miR-375 levels were significantly higher in serum (containing high-density lipoprotein (HDL) and LDL) supplemented ACM compared with serum-free ACM. In contrast, the level of miR-183-5p, which localizes to exosomes, remained unaffected by the presence or absence of serum (Fig. 3a). This pointed toward critical serum factors in stabilizing and transporting miR-375. Therefore, we isolated HDL and LDL fractions from these ACM samples following the method described by Yamamoto et al.[40] and extracted RNA for qPCR detection of miR-375. MiR-375 was exclusively present in the LDL fraction of ACM. Both LDL and HDL fractions were devoid of control candidate miR-183-5p (Fig. 3b), which is vesicle bound and RNase protected (Supplementary Fig. 4a, b). Considering the existence of miR-LDL complexes and ongoing tumor cell apoptosis, we asked whether MΦ CD36 might be involved in "miR-375-LDL" uptake. Blocking CD36 by anti-CD36 monoclonal antibody during ACM treatment (Fig. 3c) or during the coculture with MCF-7 cells (Fig. 3d) significantly attenuated the miR-375 increase in MΦ compared with an IgG isotype control. Similar results were

obtained using a CD36-blocking peptide during ACM treatment (Fig. 3e). Finally, CD36 was knocked down in MΦ by siRNA with ~70% efficiency (Fig. 3f). The resulting MΦ were subjected to our coculture protocol. The CD36 KD largely prevented the miR-375 uptake into MΦ (Fig. 3g). Taken together, these data suggest that apoptotic tumor cell-derived miR-375, bound to LDL, is taken up by MΦ via the CD36 receptor.

**Pleiotropic effects of miR-375 in MΦ and tumor cells**. The role of miR-375 in breast cancer cells with respect to immune cell interactions is unknown. However, as breast cancer cells have high abundance of miR-375, we wondered whether miR-375 might have a role in immunity. To investigate such a putative role, we first analyzed VCM and ACM from both control MCF-7 and miR-375 decoy MCF-7 cells for the amounts of CXCL10, CCL2, and CCL5, all known to exhibit chemotactic properties for different leukocyte subsets. The levels of CXCL10 and CCL5 were extremely low. However, the monocyte/MΦ chemoattractant CCL2 was highly abundant but reduced in miR-375 decoy compared with control ACM (Fig. 4a). We then measured CCL2 mRNA expression (Supplementary Fig. 5a) and its release from various viable and apoptotic breast cancer cell lines along with HMEC, which corroborated a positive correlation of miR-375 levels and CCL2 expression (Fig. 2d, Supplementary Fig. 5b). These results prompted us to investigate migration of human blood-derived CD14+ monocytes toward VCM/ACM of control and decoy MCF-7 cells in a Boyden chamber assay (Fig. 4b). Control ACM induced high monocyte migration rates, which were significantly lower in response to decoy ACM after 2 h (Fig. 4b). Next, we searched for the functional consequences of increased miR-375 in MΦ. The role of miR-375 in MΦ polarization was ruled out, as miR-375 overexpression alone or in combination with polarizing factors such as IL-4 or LPS + IFNγ failed to alter the surface protein expression of CD80, CD86, and CD206 (Supplementary Fig. 6a, b) or the gene expression of *IL1B*, *IL12*, *CLEC7A*, and *CCL18* in primary human MΦ (Supplementary Fig. 6c). Furthermore, we used a scratch assay where MΦ were treated with control or decoy MCF-7 VCM/ACM for 30 min to allow miR-375 uptake, then cultured for 24 h in fresh media to access whether miR-375 might affect MΦ migration. Interestingly, ACM from control MCF-7 cells promoted gap closure faster (~ 50%) compared with decoy ACM (~ 20%) after 24 h (Fig. 4c), which was correlated with the miR-375 content in VCM/ACM of control and decoy MCF-7 cells (Supplementary Fig. 5c, d). Furthermore, MΦ treated with MCF-7 ACM or transfected with synthetic miR-375 mimic show enhanced expression of *CCL2* (Supplementary Fig. 5e). These results suggest pleiotropic effects of miR-375 by facilitating monocyte and MΦ migration, at least in part, by induction of CCL2 in tumor cells and by unknown mechanism in MΦ.

**PXN and TNS3 are miR-375 targets in MΦ**. To understand the molecular mechanism of miR-375-mediated MΦ migration, it was necessary to identify its target genes. As miR target prediction algorithms are low in their accuracy and biochemical methods for miR target validation are technically and analytically demanding[41], we performed an Ago-RIP-Seq as previously described[42]. In brief, MΦ were transfected with synthetic miR-375 mimic or with nonspecific cel-miR-39a, and lysed 48 h later. A pan-Ago antibody (Ago-IP) or the corresponding isotype IgG control was used to immunoprecipitate Ago complexes, followed by deep sequencing. We found enhanced paxillin (*PXN*) and tensin 3 (*TNS3*) mRNA levels in MΦ overexpressing miR-375 in comparison with the IgG control and control-transfected MΦ (Supplementary Data 2 and ArrayExpress #E-MTAB-6943).

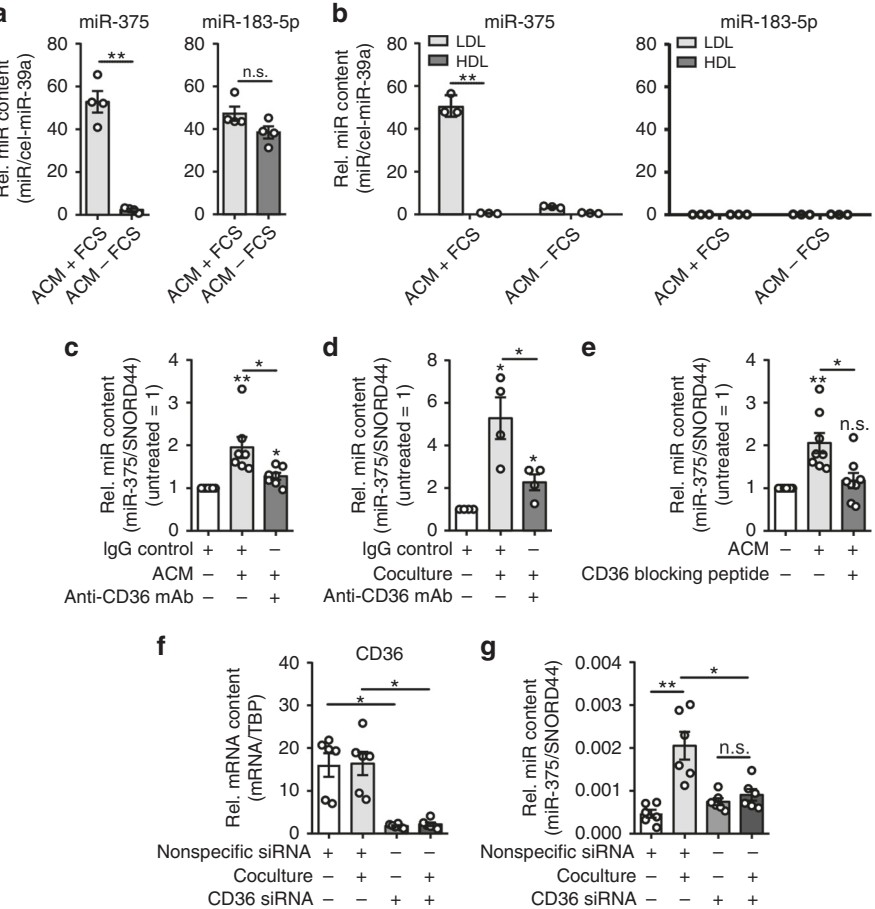

**Fig. 3** MCF-7 cell-derived miR-375 is taken up by MΦ via CD36. **a** MCF-7 cell ACM was prepared with (+ FCS) or without (− FCS) FCS, and miR-375 and miR-183-5p abundance was measured by qPCR. Synthetic cel-miR-39a was used as spike-in control for RNA purification efficiency and normalization control ($n = 4$). **b** LDL and HDL fractions were separated from ACM (with or without FCS in the media) and analyzed for the abundance of miR-375 and miR-183-5p by qPCR ($n = 3$). **c**, **d** MΦ were pre-incubated with IgG control or anti-CD36 blocking mAb for 1 h and during the whole experiment. **c** MΦ were treated with ACM for 30 min. Cells were washed and further cultured in MΦ media for 24 h. MiR-375 abundance was measured by qPCR and normalized to untreated control MΦ ($n = 7$). **d** MΦ were cocultured with MCF-7 cells for 24 h and miR-375 abundance was measured via qPCR, and normalized to untreated control MΦ ($n = 4$). **e** MΦ were treated with MCF-7 ACM alone or together with a CD36-blocking peptide for 30 min. The peptide was added 1 h before ACM treatment. Afterwards, cells were washed and further cultured in MΦ media alone or together with the peptide for 24 h. miR-375 level was measured by qPCR and normalized to untreated control MΦ ($n = 8$). **f**, **g** MΦ were transfected with control (nonspecific siRNA) or CD36 siRNA for 24 h and cocultured with MCF-7 cells for another 24 h. **f** CD36 mRNA expression and **g** miR-375 levels were measured by qPCR ($n = 6$). Data of **a**–**g** are mean ± SEM and p-values were calculated using two-tailed Student's t-test (**a**, **b**, **f**, **g**) and one-sample t-test (**c**–**e**); *$p < 0.05$, **$p < 0.01$, n.s., not significant

This suggests that the two genes might be direct targets of miR-375. These targets were then validated at mRNA and protein level in miR-375 mimic-transfected MΦ (Supplementary Fig. 7a, d, Fig. 5a), in ACM-treated MΦ (Fig. 5b), and in the coculture setting (Supplementary Fig. 7b). Protein and mRNA expression of TNS3 and PXN in ACM-treated MΦ correlated with the levels of miR-375 in control and decoy ACM (Supplementary Fig. 5d). MiRs regulate gene expression through translational inhibition and/or degradation of target mRNAs[43]. miR-375 copy numbers in ACM-treated MΦ were at par with physiological levels of endogenous miRs (Supplementary Fig. 7c)[44]. Therefore, we next analyzed PXN and TNS3 mRNA stability in MΦ overexpressing miR-375 and subsequent actinomycin D treatment to block transcription. Both PXN and TNS3 mRNA were destabilized in MΦ overexpressing miR-375 (PXN $t_{1/2} = 3.07$ h, TNS3 $t_{1/2} = 3.02$ h) as compared with controls (PXN $t_{1/2} = 10.29$ h, TNS3 $t_{1/2} = 5.68$ h) (Fig. 5c). In line with the experimental results, we detected putative binding sites of miR-375 both in PXN and TNS3 3′-untranslated regions (UTRs) (Supplementary Fig. 7e). To confirm these observations, we

transfected MΦ with vectors containing the entire 3′-UTR sequences of either PXN or TNS3, behind a Renilla luciferase coding region for 48 h. We observed a significant loss of luciferase activity when MΦ were cotransfected with miR-375 mimic (Fig. 5d). Conclusively, our results identify PXN and TNS3 as direct targets of miR-375 in MΦ.

PXN and TNS3 both have been described as regulators of cell migration[45–47]. PXN rearranges the cytoskeleton and affects the cell movement in LPS-stimulated MΦ via ERK/GSK3[48]. Its downregulation enhances endothelial cell migration[49]. Overexpression of TNS3 in HEK 293 cells inhibits cell migration, whereas downregulation in human melanoma WM793 cells increases cell migration[45]. Likewise, low levels of TNS3 in human skin fibroblasts enhance cell migration[50]. Surprisingly, not much is known about TNS3 in MΦ. We therefore posit that accumulation of miR-375 in MΦ decreases PXN and TNS3 expression to facilitate MΦ movement. To test this hypothesis, we knocked down PXN and TNS3 in human MΦ with ~ 50% efficacy, respectively (Fig. 5e), and subjected them to the scratch assay upon treatment with MCF-7 control or decoy ACM.

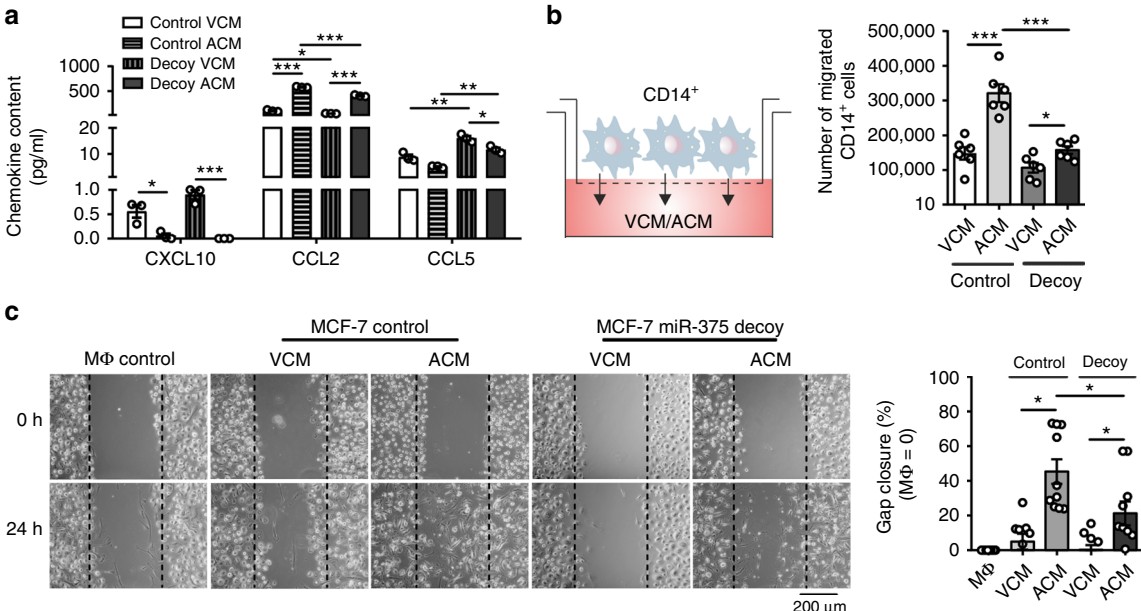

**Fig. 4** MiR-375 enhances monocyte and MΦ migration. **a** MCF-7 control or decoy VCM/ACM was analyzed for chemokine concentrations by CBA ($n = 3$). **b** $2 \times 10^6$ CD14+ monocytes were added onto transwell inserts and allowed to migrate toward empty vector-transfected (control) or miR-375 decoy-transfected (decoy) MCF-7 VCM/ACM for 2 h. The number of migrated cells in the lower chamber was calculated ($n = 6$). **c** Primary human MΦ were treated with MCF-7 control or decoy VCM/ACM for 30 min. Cells were washed and fresh MΦ media was added for 24 h. Scratches were generated with a small pipette tip in a marked area. Pictures were taken at 0 and 24 h, and the cell free area within the scratch was measured using the ImageJ software. Percentage gap closure after 24 h was calculated with respect to gap area at 0 h and normalized to untreated MΦ control (MΦ control = 0%; $n = 10$). Data of **a–c** are mean ± SEM and p-values were calculated using two-tailed Student's t-test (**a**, **b**) and two-way ANOVA with Bonferroni's correction (**c**). *$p < 0.05$, **$p < 0.01$, ***$p < 0.001$

Defective gap closure in MΦ treated with decoy ACM for 24 h was rescued by the *PXN/TNS3* double KD (Fig. 5f, g). To test the direct effect of miR-375 on PXN and TNS3 activity, we performed loss-of-function experiments using custom-designed locked nucleic acid and phosphorothioate-modified oligonucleotides (target site blockers; TSBs) that binds to the *PXN* and *TNS3* 3′-UTR to prevent miR-375 binding (Fig. 6a). Transfection of TSBs in primary human MΦ specifically rescued *PXN* and *TNS3* mRNA and protein expression that was reduced as a result of miR-375 mimic (Fig. 6b, c, Supplementary Fig. 8a) or treatment with ACM (Fig. 6d, e, Supplementary Fig. 8b). These TSBs have no effect on other miR-375 target genes, e.g., *JAK2*, *PDK1*, and *SPAG9* (Supplementary Fig. 8c). This rescue effects resulted in defective scratch closure using ACM (Fig. 6f, g). Collectively, these data suggest that tumor-derived miR-375 increased MΦ migration by downregulating the migration inhibitory proteins PXN and TNS3 in MΦ.

**Tumor-derived miR-375 is required for MΦ infiltration.** Next, we explored a more physiological setting using tumor spheroids and an in vivo model. First, we generated three-dimensional (3D) tumor spheroids from control and decoy MCF-7 cells using the liquid-overlay technique. Corroborating published findings, decoy spheroids showed reduced diameters compared with controls at 240 h (Supplementary Fig. 9a) due to reduced tumor growth in miR-375 decoy cells[21], an effect that became obvious from 168 h onwards. Hence, 4-day-old spheroids were infiltrated with CD14+ human peripheral blood monocytes for 3 days (Fig. 7a) and analyzed by flow cytometry (Supplementary Fig. 9b). MCF-7 cell numbers were unaltered but the number of infiltrated MΦ was significantly decreased in decoy spheroids by roughly ~35% (Fig. 7b). The reduction was independent of MΦ or MCF-7

apoptosis (Fig. 7c). Next, we measured 3D coculture supernatants for the abundance of chemokines that already appeared different in two-dimensional (2D) cocultures. CXCL10 and CCL2 were significantly reduced in infiltrated decoy spheroid cocultures compared with control spheroids, whereas CCL5 levels remained similar (Fig. 7d). To identify the potential source of these factors, we magnetically separated MCF-7 cells and MΦ from 3D cocultures using CD14 microbeads and analyzed mRNA expression. Expressions of *CXCL10* and *CCL5* were significantly increased in decoy MCF-7 cells, whereas *CCL2* expression was significantly decreased (Supplementary Fig. 9c). Interestingly, mRNA expression of all chemokines significantly decreased in CD14+ sorted MΦ infiltrated into decoy spheroids as compared with MΦ infiltrated into control spheroids. The mRNA induction in sorted CD14+ MΦ was manifold higher for *CXCL10* and *CCL2* as compared with tumor cells. In addition, we confirmed the reduced miR-375 transfer to MΦ in decoy spheroids (Fig. 7e), which was in line with lower miR-375 levels in MCF-7 decoy spheroid supernatants (Supplementary Fig. 9d). Furthermore, we checked mRNA expression of *PXN* and *TNS3* in MΦ separated from spheroid cocultures. In agreement with our previous data with 2D cultures (Fig. 5), the expression of both genes significantly increased in MΦ from decoy MCF-7 spheroid cocultures (Fig. 7f), which correlated with reduced miR-375 transfer. To visualize these results using a spatiotemporal method, fluorescent-labeled CD14+ monocytes infiltrated spheroids were analyzed by 3D imaging using light-sheet microscopy[51]. There was no difference in total volume of control and decoy spheroids; however, as seen in other experiments, the volume of total infiltrated labeled MΦ significantly decreased in decoy MCF-7 spheroids (Fig. 7g, Supplementary Movie 1, 2). We also measured CD14+ cell infiltration in 3D spheroids of ER + (MCF-7, T47D) and ER− (MDA-MB-468, MDA-MB-231) tumor cells (Supplementary Fig. 10a, c).

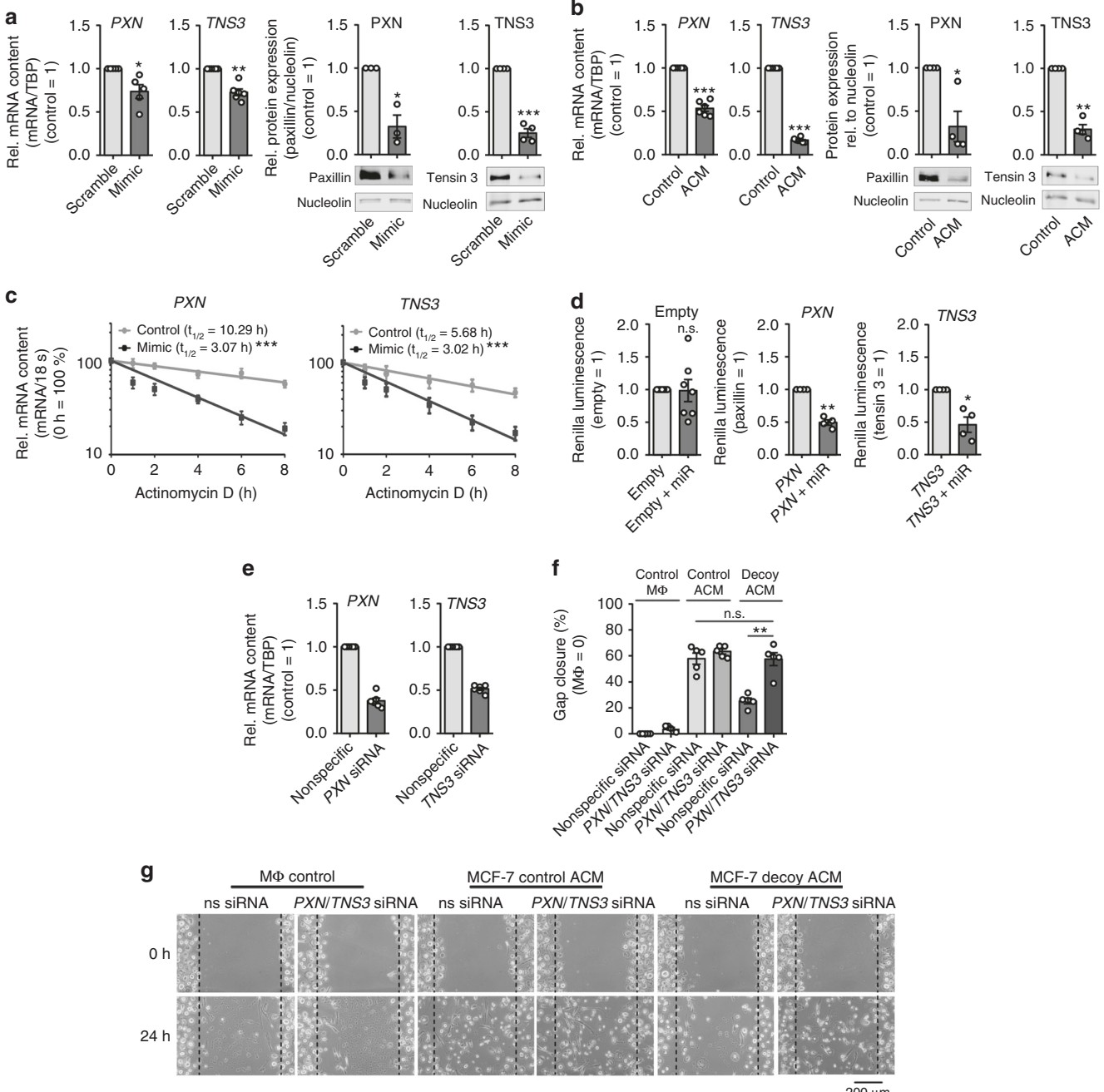

**Fig. 5** *PXN* and *TNS3* are direct targets of miR-375 in human MΦ. **a**, **c** MΦ were transfected with synthetic miR-375 mimic or cel-miR-39a control (scramble) for 72 h. **a** mRNA and protein expression of *PXN* and *TNS3* was determined. Data are normalized to transfection controls for mRNA and nucleolin for protein ($n \geq 3$). **b** Relative *PXN* and *TNS3* mRNA and protein expression in ACM-treated MΦ were normalized to control MΦ or nucleolin, respectively ($n \geq 4$). **c** mRNA stability of *PXN* and *TNS3* was measured. MΦ were treated with actinomycin D for 0–8 h. *PXN* and *TNS3* mRNA contents at the time of adding actinomycin D was set to 100%. *PXN* and *TNS3* mRNA expression in control-transfected MΦ (gray circle) and miR-375-overexpressing MΦ (black square) was measured at indicated time by qPCR ($n \geq 5$). mRNA half-life ($t_{1/2}$) was calculated by exponential regression curve. **d** MΦ were transfected with 2 μg *PXN* or *TNS3* 3′-UTR reporter plasmids or empty control vector with or without synthetic miR-375 mimic for 48 h. Binding of miR-375 to its target genes was analyzed as the ratio of renilla luciferase activity to firefly luciferase activity ($n = 4$). **e**–**g** Primary human MΦ were transfected with control or *PXN* and *TNS3* siRNA for 24 h. After 24 h, **e** MΦ were collected and *PXN* and *TNS3* expression was analyzed by qPCR or **f**, **g** treated with MCF-7 control or decoy ACM for 30 min. Cells were washed and fresh MΦ media was added for 24 h. Scratches were generated with a small pipette tip in a marked area. Pictures were taken at 0 and 24 h, and the cell-free area within the scratch was measured for percental gap closure normalized to untreated MΦ control (MΦ control = 0%) using ImageJ software ($n \geq 5$). Data of **a**–**f** are mean ± SEM and *p*-values were calculated using one-sample *t*-test (**a**, **b**, **d**, **e**) and two-way ANOVA with Bonferroni's correction (**f**). *$p < 0.05$, **$p < 0.01$, ***$p < 0.001$; n.s., not significant

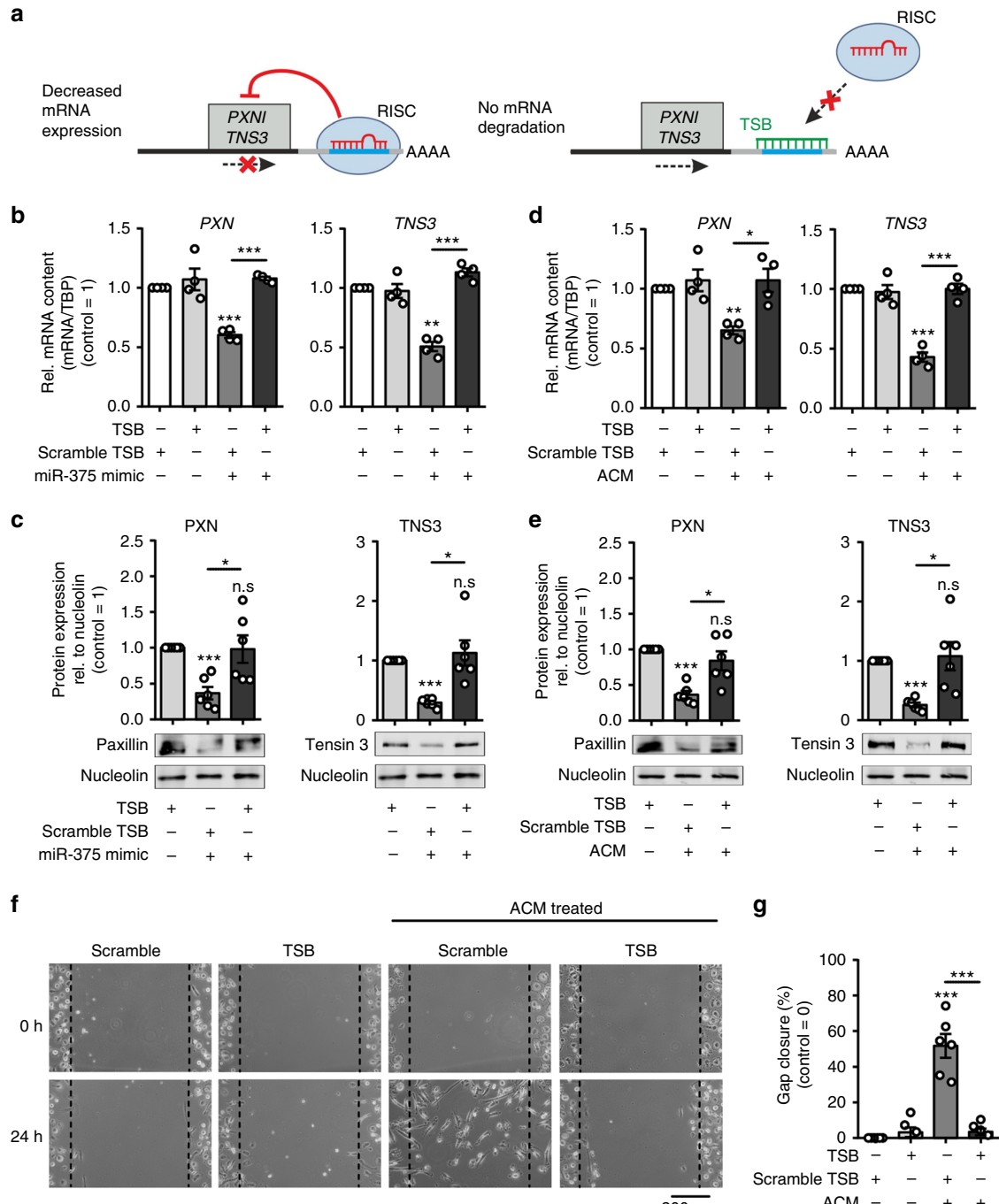

**Fig. 6** MiR-375 target site blockers for PXN and TNS3 activity. **a** Schematic overview of LNA-enhanced target site blocker (TSB) mode of action. Without TSBs, miR/RISC complex binds to the 3′-UTR of target mRNAs, thereby attenuating mRNA expression. **b**, **c** MΦ were transfected with synthetic miR-375 mimic in the presence of scramble TSBs or *PXN*- and *TNS3*-specific miR-375 TSBs for 72 h. **b** *PXN* and *TNS3* mRNA expression was measured by qPCR ($n = 4$). **c** Relative protein expression of PXN and TNS3 in MΦ. Expression was normalized to nucleolin ($n = 6$). **d**–**g** MΦ were transfected with scramble TSBs or *PXN*- and *TNS3*-specific miR-375 TSBs for 24 h. Afterwards, MΦ were treated with MCF-7 ACM for 30 min. Cells were washed and fresh MΦ media was added for another 24 h. **d** *PXN* and *TNS3* mRNA expression was measured by qPCR ($n = 4$). **e** Protein expression of PXN and TNS3 in MΦ relative to nucleolin ($n = 6$). **f** Scratches were generated with a small pipette tip in a marked area. Pictures were taken at 0 and 24 h, and the cell-free area within the scratch was measured using ImageJ software. **g** Percentage gap closure after 24 h was calculated with respect to gap area at 0 h and normalized to scramble TSB transfected MΦ (control = 0%) ($n = 6$). Data are mean ± SEM and *p*-values were calculated using one-sample *t*-test (**b**–**e**) and two-way ANOVA with Bonferroni's correction (**g**). *$p < 0.05$, **$p < 0.01$, ***$p < 0.001$, n.s., not significant

Interestingly, compared with ER⁺ spheroids, CD14⁺ cell infiltration was significantly reduced in ER⁻ spheroids (Supplementary Fig. 10b), which may point toward a secondary role of ER expression in MΦ infiltration in 3D spheroids.

To elucidate the migration inducing effect of miR-375 in vivo, control or miR-375 decoy MCF-7 cells were injected into the flanks of female NMRI-*Foxn1*ⁿᵘ mice and tumor growth was monitored three times a week for 35 days (Fig. 8a). Tumor

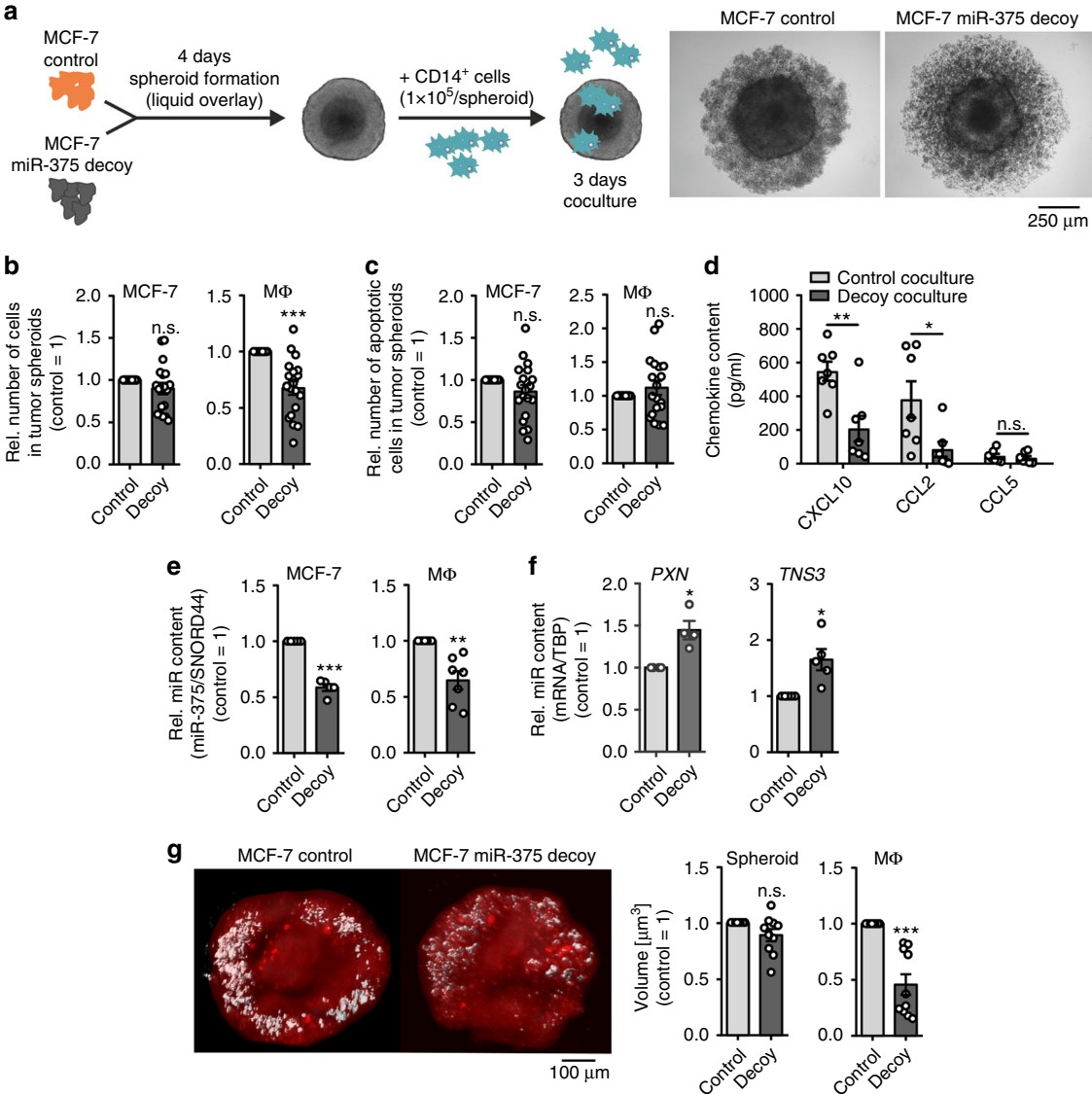

**Fig. 7** Tumor-derived miR-375 is required for MΦ infiltration in 3D tumor spheroids. **a** Schematic picture of the experimental design. Pictures were taken 3 days after coculture and are representative for 19 different experiments. **b, c** Cocultures were collected and non-infiltrating cells were removed. Single-cell suspensions of infiltrated spheroids were analyzed via polychromatic flow cytometry for **b** the number of MCF-7 cells and MΦ in tumor spheroids, and **c** the number of apoptotic cells. Cell numbers were calculated based on counting beads and normalized to control MCF-7 spheroids ($n = 19$). **d** Supernatants of cocultures were collected and chemokine concentrations were quantified by CBA ($n = 7$). **e, f** Cocultures were collected and MΦ were separated from MCF-7 cells via CD14 microbeads followed by analysis of **e** miR-375 abundance in both cell fractions and **f** *PXN* and *TNS3* mRNA expressions in MΦ by qPCR and normalized to control MCF-7 ($n \geq 4$). **g** CD14+ monocytes were stained with eFluor670 just before spheroid infiltration. Cocultures were harvested and imaged by light-sheet microscopy (× 6.3 magnification). Spheroids were analyzed for the volumes of MCF-7 cells (red) and MΦ (white), and normalized to control MCF-7 ($n = 10$ with more than 3 individual spheroids per group with 10 independent monocyte preparations). Data are mean ± SEM and p-values were calculated using one-sample t-test (**b, c, e–g**) and two-way ANOVA with Bonferroni's correction (**d**). *$p < 0.05$, **$p < 0.01$, ***$p < 0.001$; n.s., not significant

growth of decoy MCF-7 cells was retarded compared with control MCF-7 tumors (Fig. 8b), which indicates the functional relevance of miR-375 for tumor growth. After 35 days, tumors were collected, and infiltrating monocytes and MΦ were measured by fluorecence-activated cell sorting (FACS) (Supplementary Fig. 11a). In agreement with our in vitro finding, the number of infiltrated monocytes and MΦ was significantly lower in miR-375 decoy tumors compared with controls (Fig. 8c). Furthermore, the miR-375 amount was also reduced in MΦ FACS-sorted out of decoy MCF-7 tumors compared with MΦ from control tumors (Fig. 8d). Along those lines, mRNA expression of *PXN* and *TNS3* was significantly higher in MΦ from decoy tumors compared

with controls (Fig. 8e). As the mouse model comprises the interaction of human tumor cells with murine MΦ, we recapitulated experiments in the murine system. Bone marrow-derived MΦ from C57BL/6 WT mice were cocultured for 72 h with murine E0771 breast cancer cells, which also secrete miR-375. In analogy to the human system, *Pxn* and *Tns3* mRNA levels significantly decreased in primary mouse MΦ after the coculture (Fig. 8f). It is noteworthy that miR-375 mature sequences are identical in both mouse and man, and human miR-375 binds to murine *Tns3* and *Pxn* 3′-UTRs (Supplementary Fig. 11b). We then investigated in which compartment miR-375 uptake takes place and whether monocyte also take part in this process.

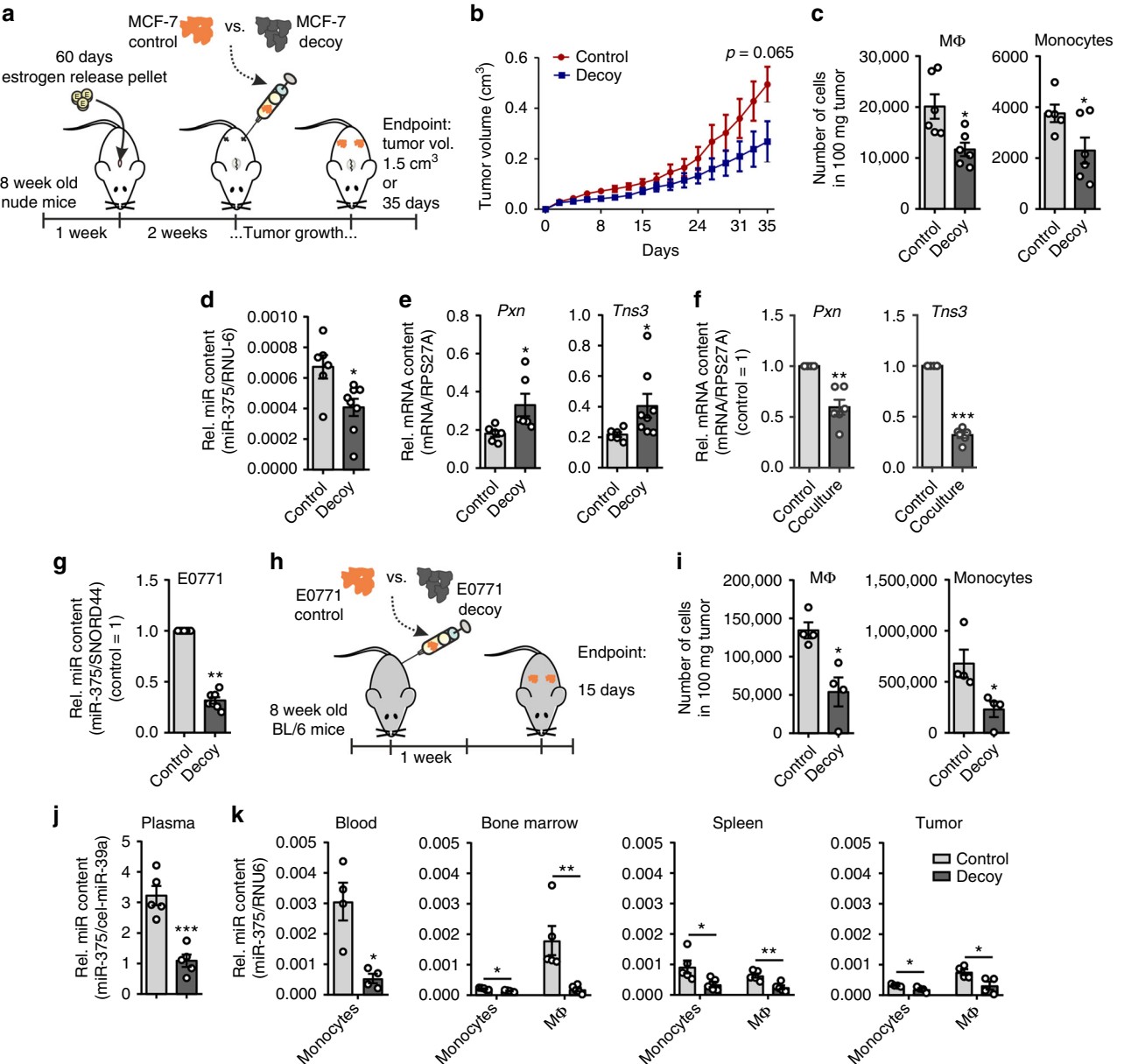

**Fig. 8** Tumor-derived miR-375 is taken up by monocytes/MΦ and facilitates their infiltration. **a–e** $1 \times 10^7$ MCF-7 control or decoy cells were injected subcutaneously in the right and left flank of female NMRI-*Foxn1[nu]* mice, which were pre-treated with 17β-estradiol pellets for 1 week. After 35 days or a maximum tumor volume of 1.5 cm³ tumors were collected for flow cytometry and cell sorting. **a** Experimental scheme. **b** The tumor growth was monitored by measuring tumor volume ($n = 5$–6 per group). **c** Tumors were analyzed for the number of infiltrating MΦ and monocytes by flow cytometry as the number of cells in 100 mg tumor ($n \geq 5$). **d, e** Infiltrating murine MΦ were sorted out of MCF-7 tumors and analyzed for the miR-375 content **d** and *Pxn* and *Tns3* expression **e** by qPCR ($n = 6$). **f** C57BL/6 mice bone marrow-derived MΦ were cocultured for 48 h with E0771 murine breast cancer cells and *Pxn* and *Tns3* mRNA expression was analyzed in MΦ ($n = 6$). **g** E0771 cells were stably transfected with miR-375 decoy (decoy) or empty vector (control) and miR-375 content was measured by qPCR ($n = 7$). **h–k** 50,000 E0771 control or decoy cells were injected in mammary gland 3 and 8 of 8-week-old female C57BL/6 mice. After 14 days, blood, bone marrow, spleen, and tumors were collected for flow cytometry and cell sorting. **h** Experimental scheme. **i** Single-cell suspensions of tumors were analyzed for the number of infiltrating MΦ and monocytes as the number of cells in 100 mg tumor. RNA from **j** plasma as well as **k** monocytes and MΦ from blood, bone marrow, spleen, and tumors were analyzed by qPCR for the abundance of miR-375 ($n = 4$–5 per group). Data are means ± SEM. *P*-values of **i–k** were calculated using nonparametric two-tailed Student's *t*-test (**b**), two-tailed Student's *t*-test (**c–e**, **i–k**) and one-sample *t*-test (**f, g**). *$p < 0.05$, **$p < 0.01$, ***$p < 0.001$

Compared with MCF-7 cells, we used a more aggressive mouse mammary carcinoma cell line E0771 with a stable KD of miR-375 using a lentiviral decoy construct or control empty vector (Fig. 8g). These cells were orthotopically transplanted into mammary gland 3 and 8 fat pad of immunocompetent C57BL/6 mice. Palpable tumors were collected after 2 weeks along with

the spleen, bone marrow, and blood (Fig. 8h). Monocytes and MΦ were FACS-sorted from these samples (Supplementary Fig. 11c) and miR-375 content was measured by qPCR (Fig. 8k). RNA isolated from the plasma of these mice was used to detect miR-375. In agreement with the nude mouse model and human MCF-7 breast cancer cells (Fig. 8a), we noticed reduced monocyte

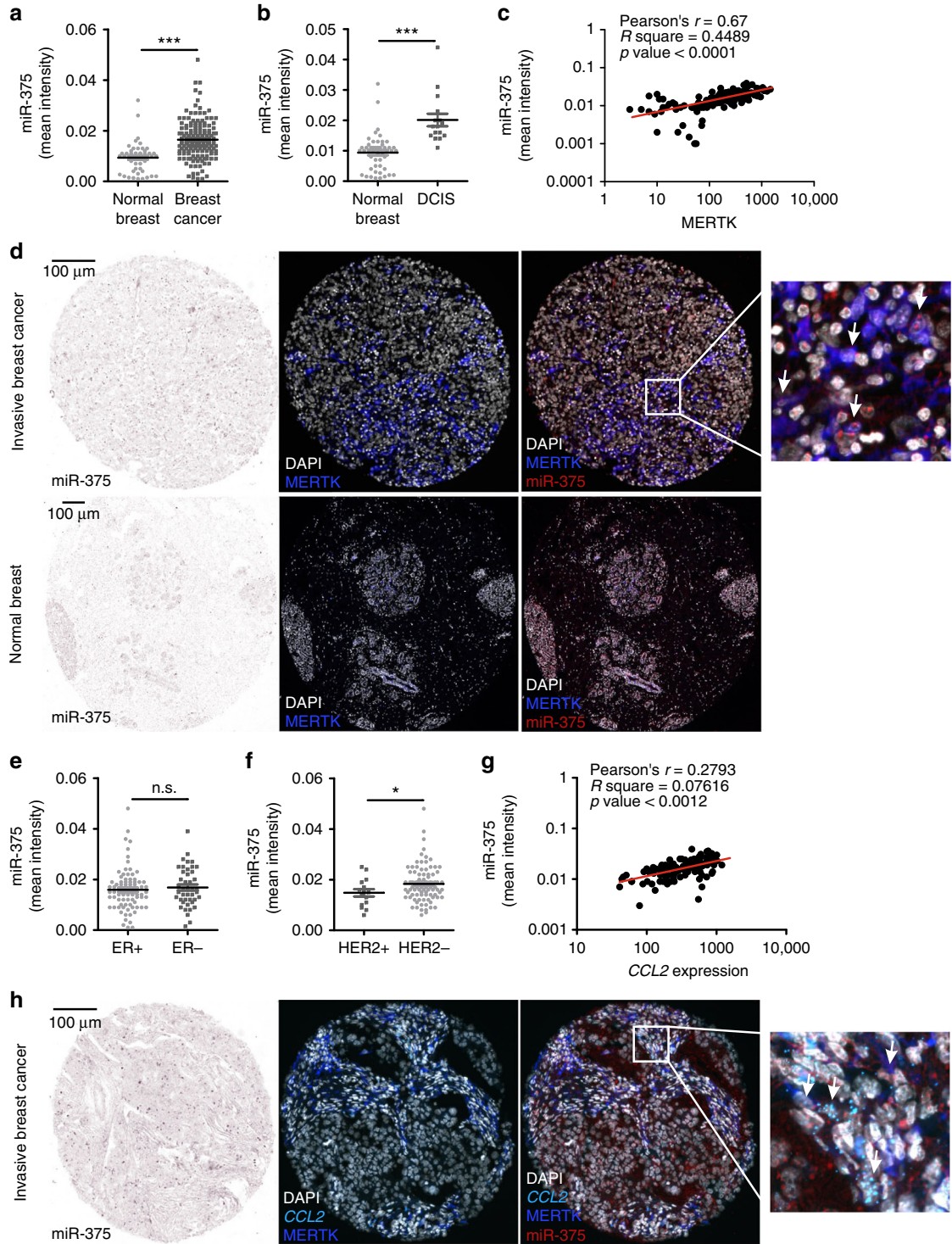

and MΦ infiltration in decoy tumors compared with control tumors (Fig. 8i). miR-375 levels in the plasma of mice receiving decoy E0771 cells were significantly lower compared with mice receiving control E0771 cells (Fig. 8j). As expected, there were reduced levels of miR-375 in blood monocytes, as well as monocytes and MΦ from decoy tumors compared with control tumors. Interestingly, we detected miR-375 in bone marrow monocytes and MΦ, with levels reflecting the situation in other organs such as the spleen, where reduced miR-375 levels were observed in cells of decoy tumor-bearing mice (Fig. 8k). Data from this mouse model suggest that the systemic release and

uptake of miR-375 by monocytes and MΦ may have far reaching consequences in addition to other tumor-derived factors. Conclusively, data from in vitro and in vivo experiments confirm that miR-375 from breast cancer cells increases monocytes/MΦ migration by targeting *PXN* and *TNS3*. Furthermore, depletion of miR-375 in breast cancer cells reduces CCL2 secretion, thereby attenuating their chemoattractive properties.

**Clinical significance of miR-375 in human mammary carcinoma.** Having identified a crucial role of miR-375 in breast

**Fig. 9** miR-375 in human invasive breast carcinoma. Human invasive mammary carcinoma tissue microarrays sections were analyzed for miR-375 abundance and *CCL2* mRNA expression by in situ hybridization (light blue), followed by staining of nuclei with DAPI (white) and MERTK protein (blue). Bright-field signal of miR-375 was converted to fluorescence image using Inform 2.4.0 and ImageJ software, to present colocalization of miR-375, MERTK, and *CCL2* from two consecutive sections. **a** Mean signal intensity of miR-375 in human invasive breast cancer (breast cancer) sections compared with normal breast tissue sections is shown ($n = 156$ breast tumors; $n = 49$ normal breasts). **b** Mean intensity of miR-375 in human ductal carcinoma in situ (DCIS) sections compared with normal breast tissue sections is shown ($n = 16$ DCIS tumors; $n = 49$ normal breasts). **c** Correlation between miR-375 mean intensity and MERTK expression in invasive breast tumor sections ($n = 155$). **d** Representative pictures of invasive breast cancer section and normal breast with arrowheads in magnification showing miR-375 (red) colocalization with MERTK (blue). **e** Mean intensity of miR-375 in ER + and ER − invasive breast tumor sections ($n = 96$ ER + and $n = 43$ ER − sections). **f** Mean miR-375 intensity of HER2 + and HER2 − invasive breast tumor sections ($n = 16$ HER2 + and $n = 89$ HER2 − sections). **g** Correlation between miR-375 mean intensity and *CCL2* mRNA expression in invasive breast tumor sections ($n = 155$). **h** Representative pictures of invasive breast cancer section and normal breast with arrowheads in magnification showing miR-375 (red) colocalization with MERTK (blue) and *CCL2* (light blue). Data are mean ± SEM and *p*-values were calculated using two-tailed Student's *t*-test. *$p < 0.05$, ***$p < 0.001$, n.s., not significant

cancer, both in vitro and in mouse models (Fig. 8), we reached out to validate the clinical relevance of miR-375 in human invasive mammary carcinoma. Datasets exploring both mRNA and miR expression of tumors and adjacent normal tissue are rare. Moreover, these datasets comprise gene expression of the whole tumor, with a limited value in accessing cell-specific gene expression. To circumvent this problem, we measured cell-specific miR-375 content using tissue microarray (TMA) slides of mammary carcinoma patients (CHTN, University of Virginia) by in situ hybridization using double digoxigenin (DIG)-labeled miRCURY LNA™ miRNA detection probes (Supplementary Fig. 12a). miR-375 staining intensity was significantly higher in invasive breast cancer compared with normal breast (Fig. 9a, d), which was also observed in ductal carcinoma in situ sections (Fig. 9b). To measure cell-specific miR-375 content, we combined miR-375 staining with the tissue MΦ marker MERTK[52,53] (Supplementary Fig. 12b) using a multispectral imaging system (PhenOptics, Perkin Elmer) ($n = 155$ patients; $n = 49$ normal breast). As presented in Fig. 9c, there was a positive correlation between miR-375 levels and MERTK expression in tumor sections with Pearson's $r = 0.67$ and $p < 0.001$. We also observed a colocalization of miR-375 and MERTK, suggesting a presence of miR-375 in human TAMs (Fig. 9d). With respect to ERα, in agreement with a previous study[21], there was no correlation with miR-375 levels (Fig. 9e). However, miR-375 levels were significantly enhanced in HER2 − samples compared with HER2 + (Fig. 9f). Next, we probed miR-375 and CCL2 correlations in TMA. As CCL2 is a secretory protein, its transcripts were quantified using RNAscope® (Supplementary Fig. 12c). After combining different detection technologies, we observed a strong correlation between miR-375 and *CCL2* (Fig. 9g), which was further substantiated by colocalization of these signals in MERTK+ TAMs (Fig. 9h). We believe that TMAs of mammary carcinoma patients substantiated our conclusion that enhanced miR-375 levels increase monocyte/MΦ migration in the tumor microenvironment and adds clinical relevance to our findings.

## Discussion
Circulating miRs serve as cancer biomarkers, also for breast cancer[54]. A study on the human plasma miR profile clearly demarcated miRs based on their association with lipoproteins or encapsulated in extracellular vesicles, including exosomes[14]. However, although dealing with the cellular source or inter-cellular miR exchange, the literature primarily focuses on exosome-mediated miR transfer or direct exchange of miRs across cells via gap junctions. The present study describes a unique transfer route of miR-375, which is not necessarily encapsulated in extracellular vesicles. miR-375 is detected at comparable levels in both exosome and exosome-depleted serum

fractions of stage II and stage III breast cancer patients[55], whereas it appears as a LDL-bound non-exosome entity in hypercholesterolemia patients[14]. Our results clearly suggest that apoptotic breast cancer cells release miR-375, as LDL bound a non-exosome entity, which is taken up by MΦ via the CD36 receptor (Figs. 2, 3). Apoptotic tumor cells also release various factors including long-chain free fatty acid and oxidized phospholipid, which may also act as a carrier of miR-375 for CD36-mediated uptake, as they are known ligands of CD36. One of the consequences of lipoprotein uptake via CD36 is fueling fatty acid oxidation by lipolysis of exogenous triacylglycerols, which has an important role in alternative MΦ activation with pro-tumoral characteristics[56]. Uptake of oxLDL by CD36 in MΦ leads to their trapping in the arterial intima and migration arrest[57], whereas, as shown here, uptake of miR-375–lipoprotein complexes provokes increased MΦ migration (Fig. 4). Interestingly, CD36 expression has been linked with poor prognosis in breast cancer and CD36 amplification specifically correlated with metastasis in a large number of human tumors, including highly aggressive melanoma[58]. Further investigations are warranted to ascertain whether miR also contributes to a tumor metastatic property via CD36 and how the CD36-mediated miR–lipoprotein complex internalization shuttles miR to the RNA-induced silencing complex to elicit their gene regulatory effects. In addition, a recent CD36-based anti-metastatic therapy[59] appears of special interest, as we propose a role of CD36 as a miR receptor.

We observed elevated levels of miR-375 only in TAMs (Fig. 1), but not in IL-4, LPS + IFNγ-polarized MΦ, or resolution-phase MΦ (Fig. 1d, Supplementary Fig. 1). This observation provides some evidence that miR-375 could be involved in modification of MΦ to display the tumor-supportive phenotypes observed for TAMs. Furthermore, the observed transfer of miR-375 from apoptotic tumor cells to MΦ raised questions toward the functional role of this miR in MΦ. Most datasets of human breast tumors comprise whole tumor "omics," whereas the availability of global gene expression profiles of stromal cells such as TAMs is very limited. Nevertheless, we tested several in silico predicted targets of miR-375, which were identified mostly in tumor cells, but those were not expressed in MΦ. Exploring an unbiased approach using Ago-RIP-Seq, we succeeded in identifying *TNS3* and *PXN* as targets of miR-375 in the human MΦ genome-wide targetome and thoroughly validated these targets in vitro and in vivo (Figs. 5–8). Evidently, miR-375 affects migration and infiltration of MΦ (Figs. 4–8). Tumor-derived miRs altering the MΦ phenotype have recently been reported, with most studies addressing changes of the MΦ polarization per se[12]. We provide information that tumor-derived miR-375 alters MΦ infiltration and migration by targeting *TNS3* and *PXN* mRNA expression, without affecting polarization (Supplementary Fig. 6). PXN is a focal adhesion adapter protein, whose activity often is regulated

via phosphorylation by Src and focal adhesion kinases[60]. The expression of *PXN* is regulated by miRs in different types of cancer[60,61]. Depending on its subcellular localization, i.e., the leading edge or the tail end of cells, it can positively or negatively regulate cell migration. Soluble tumor-derived factors, including vascular endothelial growth factor, decrease PXN expression in capillary endothelial cells[49]. In granulocyte-macrophage colony-stimulating factor (GM-CSF) generated bone marrow-derived MΦ, phosphorylation of PXN by Pyk2/FAK initiates proteolysis of PXN, and subsequent cell rounding and detachment in a CD45-dependent manner[62]. Similarly, TNS3 acts as a linker between the extracellular matrix and the cytoskeleton, and has been described as a negative regulator of cell migration in cancer cells[45,46]. Apparently, TNS3 is regulated by, and functionally contributing to, the switch between adhesive and non-adhesive states in breast cancer cells. In the light of the cellular functions of both PXN and TNS3, it appears plausible that a decreased expression of both genes increase monocyte and MΦ infiltration in 3D spheroids (Fig. 7) and in tumors from a xenograft mouse model (Fig. 8), besides reducing migration in the scratch assays (Figs. 5, 6). Intriguingly, in MΦ treated with ACM we observed a greater reduced expression of both genes as compared with the MCF-7-MΦ coculture set-up (Fig. 5b, Supplementary Fig. 7b), which cannot be solely explained by miR-375 levels. Suggestively, additive factors secreted by apoptotic breast cancer cells might trigger downregulation of adhesion proteins.

Mere downregulation of adhesion molecules by miR-375 probably might not account for the tremendously increased migration of MΦ (Fig. 5). Rather, this might also result from the reduced capacity of miR-375 decoy MCF-7 cells to recruit MΦ. Apoptotic miR-375 decoy MCF-7 cells secreted significantly lower amounts of CCL2 (Fig. 4a, Supplementary Fig. 9c), which is one of the key chemokines to regulate migration and infiltration of monocytes/MΦ[63]. Mechanisms of how miR-375 enhances *CCL2* expression are not known. However, a similar enhancer property of miR-375 has been recently reported for thymic stromal lymphopoietin induction in a human colorectal adeno-carcinoma cell line[64]. In MΦ, one possibility how miR-375 might regulate CCL2 expression is by targeting protein phosphatase 2 subunit B, which is required for activating dephosphorylation of the transcription factor FOXK1 in response to mTORC1 signaling to induce CCL2 in TAMs[65]. Interestingly, the amount of CCL5 in ACM from decoy MCF-7 cells was also higher as compared with control cell ACM (Fig. 4a). Due to differences in chemokine secretion between control and miR-375 decoy MCF-7 cells, it would be interesting to investigate migration of other immune cell subsets, such as T cells, natural killer cells, and dendritic cells. Several studies suggested a pertinent role of soluble mediators released from apoptotic cells in phagocyte attraction, known as "find-me"-signals[66,67]. Among them, lyso-phosphatidylcholine, sphingosine-1-phosphate, CX3CL1, and the nucleotides UTP and ATP are the most prominent ones. It might be interesting to examine whether miR-375 affects "find-me"-signal secretion in breast cancer cells.

Regulating tumor cell gene expression and targeting RNAs became an active area of research. The mode of miR uptake described in our study may have far reaching clinical implications as more than 70 registered clinical trials use exosomes, either as disease markers or as therapeutic molecules delivering drugs such as miRs. A notable example is a withdrawn clinical trial (ClinicalTrials.gov Identifier: NCT01344109) to evaluate tumor-derived exosomes as diagnostic and prognostic markers in breast cancer patients receiving neoadjuvant chemotherapy. The trial excluded non-exosomic miRs such as miR-375, which was found to be upregulated in drug-resistant breast cancer patients (GEO accession GSE73736). The present study not only underscores

the role of the extra-exosome regulatory molecule miR-375 but also highlights its uptake via the CD36 receptor. Understanding how cancer cells communicate with the stroma within the tumor microenvironment, especially through non-exosomic entities, may uncover an avenue for tumor-specific RNA targeting.

## Methods

**Reagents.** Nocodazole, cytochalasin B, carbenoxolone, actinomycin D, LPS, and dimethyl sulfoxide were purchased from Sigma-Aldrich (München, Germany). RNase A was procured from Qiagen (Hilden, Germany). Human IL-4, IFNγ, the monoclonal CD36-blocking antibody (#21270361; 2 µg/mL), and corresponding IgG control (#21915011; 2 µg/mL) were ordered from ImmunoTools (Friesoythe, Germany). CD36 short blocking peptide SLINKSKSSMF was synthesized according to ref. [68]. All reagents were dissolved according to the manufacturer's instructions.

**Cell culture.** Human MΦ were cultured in RPMI 1640 containing 5% AB-positive human serum (DRK Blutspendedienst Baden-Würtemberg-Hessen, Frankfurt, Germany), 100 U/mL penicillin, and 100 µg/mL streptomycin (MΦ media). MCF-7 cells were purchased from ATCC-LGC Standards GmbH (Wesel, Germany) and maintained in RPMI 1640 containing 1% sodium pyruvate and 1% non-essential amino acids, 10% FCS, 100 U/mL penicillin, and 100 µg/mL streptomycin. All cell lines were routinely tested for mycoplasma contamination and were maintained as described in Supplementary Table 2. If not stated otherwise, all cell culture supplements came from PAA Laboratories, Cölbe, Germany.

**Generation of human macrophages from buffy coats and coculture with tumor cells.** Primary human monocytes (peripheral blood mononuclear cells, PBMCs) were isolated from buffy coats of anonymous healthy donors obtained from DRK Blutspendedienst using Ficoll-Hypaque gradients (PAA Laboratories). Cells were washed twice with phosphate-buffered saline (PBS) and seeded onto high-adherence culture dishes (Sarstedt, Nümbrecht, Germany). After culturing PBMCs for 1 h in RPMI 1640 supplemented with 100 U/mL penicillin and 100 µg/mL streptomycin, non-adherent cells were washed away and remaining monocytes were cultured in media containing human serum for 7 days to allow the differentiation toward MΦ. For coculture experiments, MΦ were cultured at a density of $5 \times 10^5$ cells/mL. Tumor cells were detached from culture flasks using trypsin-EDTA and washed with PBS. Tumor cells were resuspended in MΦ media and cocultured with MΦ at the same density. After times indicated, residual MCF-7 cells were removed from the plates using trypsin-EDTA for 3–5 min, which left adherence of MΦ unaltered.

**MiR genome analysis of primary human macrophages.** Primary human monocyte-derived MΦ were cocultured with MCF-7 breast carcinoma cells for 48 h or 100 nM resolvin D1 (Cayman Chemical, Michigan, USA) for 6 h. After 48 h, residual MCF-7 cells were removed from plates and cocultured, as well as resolvin D1-treated MΦ were collected for RNA isolation using the mirVana™ miRNA isolation kit (ThermoFisher Scientific, MA, USA) according to the manufacturer's instructions. Complementary DNA libraries from total RNA were generated using the TruSeq® Small RNA sample preparation protocol (Illumina, San Diego, USA). Enrichment and size distribution of the libraries were quality-assessed by capillary electrophoresis on a DNA high-sensitive chip on Bioanalyzer (Agilent, Waldbronn, Germany). Afterwards, high-throughput DNA sequencing was performed on a NextSeq500 platform (Illumina). For bioinformatics analysis, raw sequencing reads were converted into fastq files and demultiplexed without allowing a mismatch within the barcode sequences. Adaptor clipping was performed according to TruSeq small RNA protocol. Sequencing reads were mapped using bowtie 1.0.0 against the latest release of the human genome GRChR38 and presented as normalized counts per miR (normalized for sequencing depth).

**miRseq data preprocessing.** Low-quality regions and adapter sequences from the raw sequence reads were trimmed with Trimmomatic[69]. Leading and trailing positions were clipped with a base quality value[70] below 3 followed by sliding window analysis and reads were clipped if a window of length 4 was encountered where the average base quality dropped below 15. Subsequently, adapters of the Illumina TruSeq small RNA kit and the DPN2 expression primer were screened and removed using the option ILLUMINACLIP:2:30:7 of Trimmomatic. In a second round of adapter clipping, remaining short adapter sequences were removed using the option ILLUMINACLIP:0:30:5. Sequence reads with a remaining length below 18 bp were discarded.

**Read mapping and miR identification.** Pre-processed sequence reads were mapped against the human genome GRChR38 with Bowtie[71]. An exact match of the seed was required allowing maximal one high-quality mismatch. It was considered that both reads mapping uniquely to the human genome and additionally reads with up to 100 mapping positions. Precisely, Bowtie was called with the options -n 0 -l 15 -e 40 -k 100 -m 101 --best --strata --chunkmbs 128 -t -S. The mapping results were sorted and reformatted with the samtools package[72]. Next,

for each sequence read, all of its mapping positions were compared against the genomic position of annotated human miR in miRBase release 21[73]. In the first pass, only mature miR were considered. If a sequencing read showed no overlap with mature miR, precursor miR were considered additionally in a second pass. In the case that a read with more than one mapping position in the human genome overlapped with more than one miR, each sequence read–miR assignment was weighted such that the weight across all assignments for this read sums to 1.

**Analysis of differentially expressed miRs.** The resulting miR count table was analyzed with DESeq2, version 1.18.1, and the results were represented in MA-Plots. For the heatmap display, only miR with an absolute fold change > 1.5 and a baseMean > 5 were considered. All computations were done with R package version 3.4.2.

**Generation of apoptotic cells and conditioned media.** Cells were seeded in culture flasks and cultured until they reached 90% confluency. Apoptotic cells were generated by treating cells with 1 μg/mL STS (Sigma-Aldrich) for 1 h followed by three washing steps with PBS. Cells were then used for coculture with human MΦ. To obtain ACM for RNA isolation, quantification of chemokines, or MΦ treatment, cells were further cultured in fresh media for 16 h. VCM was generated accordingly without the apoptotic stimulus. Supernatants were collected and centrifuged for 10 min at 4 °C and $2000 \times g$ to remove cells followed by centrifugation for 10 min at 4 °C and $4000 \times g$ to remove residual cellular contents and debris[74]. MΦ were subsequently stimulated with 1:1 diluted ACM/VCM in MΦ media to rule out the effect of spent media.

**Generation of stable MCF-7/E0771 miR-375 decoy and control cell line.** For production of lentiviral particles, $1.5 \times 10^6$ HEK293T cells were seeded in 10 cm culture dishes and transfected with lentiviral vector encoding miR-375 decoy insert (Addgene, Cambridge, USA) or empty vector using jetPRIME® Transfection Reagent (Polyplus, New York, USA) according to the manufacturer's instructions. As packaging plasmids, psPAX2 and pMD2.g (both from Addgene) were used. Twenty-four hours after transfection, fresh media was added for additional 24 h. Supernatants containing lentiviral particles were collected, centrifuged at $500 \times g$ and room temperature for 5 min to remove cells, and filtered through 0.22 μm filters. MCF-7 or E0771 cells ($3 \times 10^5$) were seeded per well in six-well plates and treated with lentiviral particle-containing media for 24 h. The next day, fresh media was added. Transduction efficiency was analyzed based on green fluorescent protein signal in the cells with LSRII/Fortessa flow cytometer (BD Bioscience, Heidelberg, Germany).

**RNA isolation, reverse transcription, and quantitative real-time PCR.** Isolation of RNA from cells and cell culture supernatants was performed using PeqGold® (Peqlab Biotechnologie, Erlangen, Germany) according to the manufacturer's instruction. For miR isolation from supernatants, 5 pg of synthetic miR-39a from *Caenorabditis elegans* (cel-miR-39a; Qiagen) were added as a spike-in control for purification efficiency[75]. RNA was transcribed into cDNA for mRNA analysis using Fermentas' reverse transcriptase Kit (ThermoFisher Scientific) and for miR analysis using MystiCq™ microRNA synthesis mix (Sigma-Aldrich) according to the manufacturer's instructions. Real-time qPCR was performed using the CFX96™ Real-Time PCR Detection System and SYBR green (both from Bio-Rad Laboratories, München, Germany). Primers for hsa-miR-375, hsa-miR-183-5p, hsa-miR-21-5p, hsa-miR-511-3p, hsa-miR-142-3p, hsa-miR-33a, hsa-let-7a-5p, and SNORD44 were from Sigma-Aldrich. Primer for cel-miR-39a was from Qiagen. All other primers were from Biomers (Ulm, Germany) and sequences are presented in Supplementary Table 1. Relative mRNA/miR expression was calculated using the CFX-Manager™ v3.2 software (Bio-Rad Laboratories) and the $\Delta\Delta C_t$ method, and normalized to respective control RNAs indicated. For absolute quantification, synthetic cel-miR-39a was used to generate a standard curve of Ct values (y-axis) against log copy number (x-axis) according to Qiagen's instructions.

**Tumor spheroid generation and coculture with primary human monocytes.** To generate 3D tumor spheroids from MCF-7 and T47D cells, the liquid-overlay technique was used as described[76]. Briefly, $5 \times 10^3$ cells per well were seeded onto non-adherent 1% agarose-coated 96-well plates and allowed to form spheroids for 4 days. To initiate spheroids from MDA-MB-231 and MDA-MB-468 cells, $5 \times 10^3$ cells were seeded per well onto 96-well round-bottom plate and centrifuged for 15 min at $1000 \times g$. Five percent Matrigel (Corning, NY, USA) in culture media was added to a final concentration of 2.5% per well and cells were allowed to form spheroids for 4 days. Primary human monocytes were isolated from human blood PBMCs by using CD14 microbeads (Miltenyi Biotec, Gladbach, Germany) and the AutoMACS Separator system (Miltenyi Biotec). Monocytes ($1 \times 10^5$) were added per spheroid and cocultures were maintained for 3 days to allow monocyte infiltration. For light-sheet fluorescence microscopy of tumor spheroids, CD14+ cells were stained with cell dye eFluor670 (eBioscience, Frankfurt, Germany) according to the manufacturer's protocol before coculture. To determine spheroid diameters, pictures of spheroids were taken using a Canon EOS 600D camera (Canon, Krefeld, Germany) and a transmitted-light microscope (AxioVert 40; Zeiss, Oberkochen, Germany) at ×5 magnification. Pictures were analyzed using ImageJ software.

**Animal models.** Mouse care and experiments involving mice were approved by and followed the guidelines of the Hessian animal care and use committee (approval number FU/1152). For xenograft transplantation experiments of MCF-7 control or MCF-7 miR-375 decoy cells, 8–12 weeks old female NMRI-Foxn1nu mice (Charles River laboratories, Sulzfeld, Germany) were used. One week before tumor cell injection, 17β-estradiol pellets (1.5 mg/pellet, 60 days release; Innovative Research of America, Sarasota, USA) were implanted into the mice between the ear and shoulder. MCF-7 ($1 \times 10^7$) control or decoy cells per site were subcutaneously injected at the right and left flank of the mice. Tumor growth was monitored by measuring the length ($L$) and the width ($W$) using sliding calipers and the volume (V) was calculated according to the formula $V = 0.5 \times (L \times W^2)$. After 35 days, animals were perfused and tumors were collected for flow cytometry and cell sorting. To investigate miR-375 transfer to monocytes and MΦ of different organs, 50,000 E0771 miR-375 decoy or control, cells were injected into mammary gland 3 and 8 of 8-week-old female C57BL/6 mice. After 14 days of tumor growth, animals were killed and the blood, spleen, bone marrow, and tumors were collected for cell sorting.

**Immunoblotting.** MΦ were collected in lysis buffer (containing 6.65 M urea, 10% glycerol, 1% SDS, 10 mM Tris pH 6.8; pH 7.4) and incubated for 5 min at 95 °C with $1 \times$ SDS sample buffer ($5 \times$ buffer consists of 100 mM Tris-HCl pH 6.8, 5% SDS, 25% glycerol, 0.01% bromphenol blue, 100 mM dithiothreitol) and resolved on polyacrylamide gels followed by transfer onto nitrocellulose membranes. Nonspecific binding was blocked with 5% milk powder in Tris buffer saline with 0.05% Tween-20 for 1 h at room temperature, followed by incubation with primary antibodies against PXN (Abcam, Cambridge, UK; #ab32084; 1:3000 dilution), TNS3 (ThermoFisher Scientific #PA5-63112; 1:750 dilution), and nucleolin (Santa Cruz Biotechnology, Heidelberg, Germany; #sc-8031; 1;3000 dilution) according to the manufacturer's instructions. Proteins were visualized with IRDye secondary antibodies using the Li-Cor Odyssey imaging system (all from LICOR Bioscience, Bad Homburg, Germany). Uncropped blots are presented in Supplementary Figures.

**HDL and LDL separation.** HDL and LDL fractions from MCF-7 cell ACM were separated using the LDL/VLDL and HDL purification kit (Cell Biolabs, Inc., San Diego, CA) following the manufacturer's instructions and as described[40]. In brief, lipoprotein complexes were precipitated via dextran sulfate and further purified through additional precipitation and low-speed differential centrifugation. As a control, ACM without additional FCS was used for lipoprotein isolation.

**In situ hybridization and multiplex immunohistochemistry.** TMAs of human normal breast and invasive breast cancer were provided by the Cooperative Human Tissue Network and the Cancer Diagnosis Program, which are funded by the National Cancer Institute. Other researchers may have received exemplars from the same subjects. In situ hybridization of double DIG-labeled miRCURY LNA™ miRNA Detection Probe hsa-miR-375 (Qiagen, #YD00610232) and scamble-miR (Qiagen, #YD00699004) has been performed according to the miRCURY® LNA® miRNA detection probes handbook with the following modifications: sections were deparaffinized and treated with $H_2O_2$ followed by antigen retrieval and protease treatment according to the RNAscope® Multiplex Fluorescent v2 assay kit (Advanced Cell Diagnostics (ACD), Newark, USA) protocol. Hybridization of 40 nM miR-375 and scramble-miR probe was performed at 55 °C for 1 h. In situ hybridization of human *CCL2* mRNA by RNAscope® technique was carried out according to the manufacturer's instructions (ACD). HS-CCL2 (ACD, #42381) and negative control probe (ACD, #320871) were hybridized for 2 h followed by three amplification steps. The signal was detected with RNAscope® Multiplex fluorescent Detection reagent kit v2 (ACD, #323110) using Opal dyes (Perkin Elmer, Rodgau, Germany). Following in situ hybridization, sections were stained with antibody against human MERTK (Abcam, #52968; 1:2000 dilution) using the Opal staining system according to the manufacturer's instructions (Perkin Elmer). Nuclei were counterstained with 4′,6-diamidino-2-phenylindole and slides were mounted with Fluoromount-G (Southern Biotech, AL, USA). Slides were imaged at using Vectra3 automated imaging software and images were analyzed using inForm2.0 (Perkin Elmer) and ImageJ Software.

**Flow cytometry.** Single-cell suspensions were stained with fluorochrome-conjugated antibodies and analyzed on a LSRII/Fortessa flow cytometer or sorted using a FACSAria III cell sorter (both from BD Biosciences, Heidelberg, Germany). All antibodies and secondary reagents were titrated to determine optimal concentrations. CompBeads (BD Bioscience) were used for single-color compensation to create multicolor compensation matrices. For gating, fluorescence minus one controls were used. As an internal counting standard flow-count fluorospheres (Beckman Coulter, Krefeld, Germany) were applied. The instrument calibration was controlled daily using Cytometer Setup and Tracking beads (BD Biosciences). Data were analyzed using FlowJo software V10 (Treestar, Ashland, OR, USA).

To analyze MΦ phenotypes, cells were washed with PBS and pelleted at $500 \times g$ and 4 °C for 5 min. Afterwards, nonspecific antibody binding to FC-gamma receptors was blocked using 2% human Fc Receptor block (eBioscience, Frankfurt, Germany) in PBS for 15 min on ice. Cells were incubated with a mixture of

CD80-APC (BioLegend, #305220; 1:50 dilution), CD86-FITC (BD Bioscience, #555657; 1:50 dilution), CD206-PE-Cy5 (BioLegend, #321108; 1:50 dilution), CD163-PE (BD Bioscience, #558018; 1:50 dilution), and HLA-DR-PE-Cy7 (BD Bioscience, #560651; 1:200 dilution) antibodies for 20 min on ice and in the dark.

For analysis of infiltrated spheroids, cocultures were collected and washed with PBS to remove non-infiltrating monocytes, followed by treatment with accutase (PAA Laboratories, Cölbe, Germany) for 30 min at 37 °C, to obtain single-cell suspensions. Cells were washed with PBS and pelleted at $500 \times g$ and 4 °C for 5 min. Afterwards, cells were blocked with 2% Fc Receptor block (eBioscience) in PBS for 15 min on ice. Cells were incubated with a mixture of CD45-PE (BioLegend, #368509; 1:50 dilution), CD14-APC-H7 (BD Bioscience, #560180; 1:100 dilution), and CD11c-V450 (BD Bioscience, #560370; 1:200 dilution) antibodies together with Annexin V-APC (ImmunoTools, Friesoythe, Germany, #31490016; 1:100 dilution) and 7-AAD-PE-Cy5 (BD Bioscience, #559925, 1:100 dilution) to discriminate living, apoptotic, and necrotic cells, for 20 min on ice in the dark.

For FACS analysis and FACS sorting of MCF-7 tumors from xenograft transplantation experiments, single-cell suspensions were created using the human tumor dissociation kit and the GentleMACS isolator (Miltenyi Biotec, Gladbach, Germany). Single-cell suspension from murine E0771 tumors were prepared using the murine tumor dissociation kit (Miltenyi). Blocking of tumor-, blood-, spleen-, and bone marrow single-cell suspensions was performed by using 2% of human and/or murine Fc Receptor block (eBioscience), followed by staining with an antibody mixture of CD11b-eFluor605 (BioLegend, #101257; 1:200 dilution), F4/80-Pe-Cy7 (BioLegend, #123114; 1:200 dilution), Ly-6G-APC-Cy7 (BioLgened, #127624; 1:100 dilution), CD326-PE (BioLegend, #324205; 1:100 dilution), CD11c-BV711 (BD Bioscience, #363048; 1:200 dilution), Ly-6c-PerCP-PE-Cy5.5 (BD Bioscience, #560525; 1:200 dilution), CD45-VioBlue (Miltenyi Biotec, #130102430; 1:50 dilution), and HLA-DR-APC (Miltenyi Biotec, #130102139; 1:50 dilution). Cell suspensions were filtered through 30 μm cell strainer and diluted to ideal concentrations for flow cytometry and cell sorting.

**Plasmid construction.** To generate plasmids containing miR-375-binding sites for human PXN or TNS3, psiCHECK 2™-vector (Promega, Madison, USA) was digested with NotI and XhoI restriction enzymes (New England Biolabs, Frankfurt, Germany). 3′-UTRs of either PXN or TNS3 were amplified from human cDNA with primers for PXN sense 5′-TAGGCGATCGCTCGAGGTGCCCTG CCCCTGTCTC-3′ and antisense 5′-TTGCGGCCAGCGGCCGCTGAAAATCA TGGGCAAACTTT-3′, and TNS3 sense 5′-TAGGCGATCGCTCGAGGAACT CCCCTCCCTCCCT-3′ and antisense 5′-TTGCGGCCAGCGGCCGCTTACC AAGTTCATAATTTTTATTAT-3′ (all from Biomers), and inserted into linearized psiCHECK™-2 vectors with the In-Fusion® HD Cloning Kit (Takara, Frankfurt, Germany) according to the manufacturer's protocol.

**Luciferase reporter assay.** For luciferase activity assay, human MΦ cultured in six-well plates were transiently cotransfected with 2 μg PXN/TNS3 3′-UTR reporter plasmids or an empty control plasmid with or without MISSION® hsa-miR-375 Mimic (Sigma-Aldrich) using ViromerRED transfection reagent (Lipocalyx, Halle, Germany) according to the manufacturer's instructions. After 48 h, cells were lysed in 240 μL per well Passive Lysis Buffer (Promega) and firefly and Renilla luciferase activities were measured in cell lysates using the Dual Luciferase kit assay (Promega) on a Mithras LB 940 luminometer (Berthold Technologies, Bad Wildbad, Germany). Renilla luciferase activity was normalized to firefly luciferase activity in the lysates. The activity in miR-375 cotransfected cells was expressed as fold change compared with the cells transfected with vectors only.

**Isolation and culturing of bone marrow-derived macrophages.** Bone marrow was isolated from the tibia and femur of C57BL/6 mice and $6 \times 10^6$ bone marrow cells were incubated in RPMI 1640 containing 10% FCS, 100 U/mL penicillin, 100 μg/mL streptomycin, 20 ng/mL macrophage colony-stimulating factor, and 20 ng/mL GM-CSF (Peprotech, Hamburg, Germany) for 7 days. Medium was replaced every 2 days.

**Chemokine quantification.** CXCL10 (BD Bioscience, #558280), CCL2 (BD Bioscience, #558287), and CCL5 (BD Bioscience, #558324) in tumor cell VCM/ACM and tumor spheroid-monocyte coculture supernatants were quantified using cytometric bead array Flex Sets. Samples were acquired by flow cytometry and processed with FCAP software V1.0.1 (BD Bioscience).

**Spheroid clearing and light-sheet fluorescence microscopy.** To prepare MCF-7 spheroid-monocyte cocultures for light-sheet fluorescence microscopy, spheroids were collected and non-infiltrated monocytes were removed by three washing steps with PBS. Fixation of spheroids was carried out using 4% paraformaldehyde (Merck, Darmstadt, Germany) for 2 h at room temperature followed by washing with PBS. Spheroids were embedded in 1.3% low-melting agarose (Sigma-Aldrich) and dehydrated in an ascending ethanol series (30, 50, 70, 90, 96%) for 30 min each, and finally immersed twice for 30 min in 100% ethanol, in order to achieve a complete dehydration. Then, spheroids were transferred to dibenzyl ether (Sigma-Aldrich) for clearing and 3D image acquisition was done on Ultramicroscope II (LaVision BioTec, Bielefeld, Germany). Pictures were taken with a Neo 5.5 (3-tap)

sCOMs camera (Andor, Mod.No.: DC-152q-C00-FI) and the ImSpectorPro software version 5.0.110 (LaVision BioTec) with × 6.3 magnification. Z-planes were generated with 2 μm spacing. The following laser lines and filter sets were used: for MCF-7 cell imaging, spheroids were excited at 545/30 nm and detected at 595/40 nm. Monocytes stained with eFluor670 (eBioscience) were excited at 630/30 nm and detected at 690/30 nm. 3D images and quantification were done with Imaris version 7.6 (Bitplane, Zurich, Switzerland).

**miR mimic and siRNA transfection of human macrophages.** Both miR mimic and siRNA transfection were performed using HiPerfect (Qiagen) according to the manufacturer's instructions. For overexpression of miR-375 primary human MΦ in six-well plates were transfected with MISSION® hsa-miR-375 mimic or MISSION® miR negative control 2 from C. elegans (cel-miR-39a; both from Sigma-Aldrich). De novo synthesis of mature miRs in MΦ upon coculture was prevented using DICER siRNA or control siRNA (Santa Cruz biotechnology). To block CD36 gene expression, MΦ were transfected with ON-TARGETplus CD36 siRNA or control siRNA (both from Dharmacon, Lafayette, Colorado, USA). Double KD of PXN and TNS3 was performed in primary human MΦ using PXN/TNS3 siRNA or control siRNA (Santa Cruz biotechnology). To interfere with the binding of miR-375 to its targets, miRCURY LNA™ miRNA Power Target Site Blockers (Qiagen) with the following custom sequences were used according to the manufacturer's instructions: PXN 5′-CTGTCCATCCCGCACCAGCG-3′, TNS3 5′-CTCGCCCAG CTCGCCCCA-3′, and scramble control 5′-ACGTCTATACGCCCCA-3′.

**Migration assays.** For random migration scratch assays with human MΦ in six-well plates, cells were treated with MCF-7 control or decoy VCM/ACM for 30 min and washed three times with PBS before fresh MΦ media was added. Scratches were applied with a small pipette tip in a marked area and scratches were acquired using Canon EOS 600D camera (Canon) and a transmitted-light microscope (AxioVert 40; Zeiss, Oberkochen, Germany). The cell-free area within the scratch was calculated using ImageJ software. For migration assays of human monocytes toward MCF-7 control or decoy VCM/ACM, primary human monocytes were isolated from human blood PBMCs by using CD14 microbeads and the Auto-MACS Separator system (Miltenyi Biotec). Cells were washed with PBS and $2 \times 10^6$ cells were added onto Transwell inserts (6.5 mm Transwell with 5.0 μm pores; Corning, NY, USA). Monocytes were allowed to migrate for 2 h. Cells were collected from the lower chamber and the number of cells was determined using Neubauer counting chamber (Labor Optic, Friedrichsdorf, Germany).

**Ago immunoprecipitation.** To identify direct targets of miR-375, $10 \times 10^6$–$20 \times 10^6$ MΦ were treated as follows: cells were transfected with synthetic miR-375 mimic (mimic) or with nonspecific cel-miR-39a (control) for 48 h. Twenty-four hours after transfection, both control and mimic were treated with 1:1 diluted MCF-7 ACM for 30 min and cultured in fresh media for 24 h. Ago immunoprecipitation (Ago-IP) was performed as described[42,77]. Briefly, MΦ were washed with ice-cold PBS, scraped in 5 mL PBS, and pelleted at $500 \times g$ and 4 °C for 5 min. Cells were lysed in polysome lysis buffer for 5 min on ice, pooled from three individual donors, and were frozen at − 80 °C. Anti-pan-Ago IgG 2A8 (Merck Millipore, #MABE56; 10 μg) or IgG1 isotype control (ImmunoTools, #21275511; 10 μg) were coupled to protein G Dynabeads (ThermoFisher Scientific) in 500 μL NT2 buffer overnight rotating at 4 °C, followed by three washing steps with NT2 buffer. Thawed cell lysates were centrifuged for 30 min at $17.00 \times g$ and 4 °C and added 1:10 to NT2 buffer. The lysate was equally distributed to two tubes containing pan-Ago-IgG- or IgG1-coated beads and incubated overnight rotating at 4 °C. The next day, beads were collected at the magnetic part and washed five times with 1 mL NT2 buffer. Protein–RNA complexes were eluted from the beads by incubation with 50 μL glycine (pH 2.3) for 15 min at room temperature and the obtained solution was neutralized using 5 μL Tris-HCL (pH 8.0). Proteins were digested by adding 3 μL of proteinase K (Sigma-Aldrich) and incubation for 10 min at 55 °C. Thereafter, 350 μL PeqGold RNA Pure solution (Peqlab Biotechnologie) were added and RNA was purified using miRNeasy Mini Kit (Qiagen) without on-column DNase I digestion according to the manufacturer's recommendations.

**Library generation and Ago-RIP-Seq.** Ten nanograms of purified RNA samples from Ago and IgG1 fractions were DNase I (Qiagen) treated and sequencing libraries were generated by using the SMARTer Stranded Total RNA-Seq Kit–Pico Input Mammalian (Takara Clontech) according to the manufacturer's protocol. For sequencing, Ago-IP and IgG-IP fractions were pooled to generate a final concentration of 1 nM per lane. The samples were sequenced as single-end reading of 51 bases with dual-indexing of 8 index reads on a NextSeq500 platform (Illumina). Sequencing data were converted into fastq files and demultiplexed without allowing a mismatch within the adaptor sequences. Sequencing reads were mapped using SeqMan NGen (DNAstar 12.2, WI, USA) against the latest release of the human genome GRChR38. Assembled sequences were loaded into Qseq (DNAstar) to compute raw reads per kilobase of exon model per million mapped (RPKM) reads normalized expression values of the transcript isoforms. A stringent filtering criterion of RPKM value 1.0 in at least one sample was used to obtain expressed transcripts. Expression data was quantified as Ago-IP$_{IgG}$: (Ago.miR-375–IgG.miR-375) – (Ago.control–IgG.control) where miR-375 targets: log2FC (Ago-IP$_{IgG}$) > 0.

**Statistical analysis**. All data are presented as means values ± SEM of at least three independent experiments. Statistical analyses were performed using two-tailed Student's $t$-test, one-sample $t$-test, and two-way analysis of variance with Bonferroni's correction, as indicated in the figure legends. Asterisks indicate significant differences between experimental groups (*$p < 0.05$, **$p < 0.01$, and ***$p < 0.001$).

**Reporting Summary**. Further information on experimental design is available in the Nature Research Reporting Summary linked to this article.

## Data availability

miRseq and Ago-RIP-seq data are deposited in the ArrayExpress database at EMBL-EBI under the accession numbers E-MTAB-6885 [https://www.ebi.ac.uk/arrayexpress/experiments/E-MTAB-6885/] and E-MTAB-6943 [https://www.ebi.ac.uk/arrayexpress/experiments/E-MTAB-6943], respectively. All relevant data are within the paper and its Supplementary Information files. Additional data that support the findings of this study are available from the corresponding authors upon reasonable request.

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

## Acknowledgements

We thank Gudrun Bayer and Praveen Mathoor for their excellent technical assistance. We thank Jeremy Epah for assistance with light-sheet microscopy and Anica Scholz for Ago-RIP-Seq raw data curation. We acknowledge the support of Cadio-Pulmonary Institute (CPI), EXC 2026, Project ID: 390649896. The study is supported by Deutsche Forschungsgemeinschaft grants DFG SFB 1039 TP B04 (B.B.) and B06 (A.W.).

## Author contributions

Conceptualization: S.N.S. and B.B. Methodology: A.-C.F., S.E., A.F.F., S.L., A.W., T.S., I.E. and S.N.S. Formal analysis: A.-C.F., S.E. and S.N.S. Investigation: A.-C.F., A.F.F. and S.N.S. Resources: I.E. and B.B. Writing–original draft: A.-C.F. and S.N.S. Writing–review and editing: all authors. Visualization: S.N.S. and B.B. Supervision: S.N.S. and B.B. Funding acquisition: B.B.

## Additional information

**Competing interests:** The authors declare no competing interests.

