## [Peer Review File · Nature Communications]

Reviewers' comments:

Reviewer #1, Expertise: apoptosis, TME (Remarks to the Author):

This is a rigorous and robust study which demonstrates the importance of miR-375 in altering the phenotype of TAMs in breast cancer and suggesting that transfer of miR-375, especially from apoptotic tumor cells to monocytes and macrophages can profoundly influence their migratory and activation states to promote tumor growth. The authors dissect underlying mechanisms through in vitro and in vivo approaches and demonstrate (1) a molecular mechanism (macrophage CD36) for uptake of miR-375 which they prove to be non-vesicular, and (2) miR-375 target genes (macrophage PXN and TNS3). Of course this type of discovery research generates many additional questions - such as those raised by the authors in their Discussion - but the level of detailed new knowledge elucidated in this paper warrants publication after attention to minor issues, in this referee's opinion.

Minor points:

1. MCF-7 cells, on which the authors' conclusions are largely based, are unusual in that they are deficient in the effector caspase-3 which may result in subtly different apoptotic features than caspase-3-expressing tumor cells. Were the same observations on production and transfer of miR-375 seen with ACM from caspase-3+ tumor cells?
2. Is CCL2 production increased in macrophages as a consequence of miR-375 activity?

Reviewer #2, Expertise: miRNA, cancer (Remarks to the Author):

In the manuscript by Frank et al. the authors report that apoptotic breast cancer cells, independent of exosomes, release miR-375, which is taken up by tumor-associated macrophages (TAM) via the CD36 scavenger receptor. In TAM, miR-375 is accumulated and regulates TNS3 and PXN gene expression to promote infiltration into the tumor spheroid. In tumor cells, miR-375 regulates CCL2 through an indirect mechanism to recruit macrophages. The work is original and provides new understandings on the regulation of macrophage infiltration and the interaction between tumor and immune cells via extracellular RNA. The findings are novel and experiments are well designed. Data are of high quality and support the conclusion. Statistic methods are appropriate. However the following concerns need to be addressed.

1. It was previously reported that miR-375 expression level is associated with ER status in breast cancer (Cancer Res. 2010; 70:9175-84). In addition to T47D, MCF-7, and MDA-MB-231, more breast cancer cell lines (including both ER+ and ER-) and primary breast epithelial cells should be examined for their function in transferring miR-375 and recruiting macrophages in order to determine if this effect is specific to ER+ tumors.
2. Primary breast tumors should be examined for a correlation between miR-375 level and macrophage markers. This should also be compared between ER+ and ER- tumor.
3. The protein levels of PXN and TNS3 are never shown in the manuscript and should be added.
4. To confirm PXN and TNS3 indeed mediate miR-375's effect the authors should use PXN and TNS3 expression vectors with mutated miR-375 binding sites for restoration assay.
5. The mechanism through which miR-375 induces CCL2 is not identified. Since ER+ breast cancer cells express more miR-375, it would be important to test if they also produce more CCL2.
6. miR-183-5p also seems to be transferred from breast cancer cells to macrophages. What's its function? At the very least, does it synergize with miR-375 in stimulating and recruiting macrophages?

Reviewer #3, Expertise: TAMs (Remarks to the Author):

In this manuscript, Frank and coworkers show that breast cancer cells transfer miR-375 to macrophages to promote their infiltration to tumors. The authors show that miR-375 is transferred to macrophages either alone or in complex with LDL, but not in association with extracellular vesicles. According to the authors, macrophages internalize miR-375 through CD36. The authors then show that cancer cell-derived miR-375 downregulates the expression of PXN and TNS3 in macrophages. PXN and TNS3 are two novel target genes of miR-375. Furthermore, miR-375 enhances CCL2 production in cancer cells. In a tumor mouse model of breast cancer, knockdown of miR-375 in cancer cells using a decoy strategy caused Pxn and Tns3 upregulation in tumor-associated macrophages (TAMs) while decreasing macrophage and monocyte infiltration.

The manuscript is well written and the findings are novel and interesting. However, there are some aspects that are still at an early stage and should be further developed by the authors before this manuscript is considered for publication.

Comments

1. The authors claim that cancer cell-derived miR-375 modulates gene expression in macrophages upon its internalization. Previous studies have demonstrated that only miRNAs expressed at a relevant level can modulate gene expression (Brown et al., Nature Biotechnology 2007). In order to demonstrate that exogenous miR-375 can reach functional levels in macrophages, the authors should quantify the absolute expression level (number of copies) of miR-375 compared to other miRNAs, which are relevant to macrophage biology, such as miR-21-5p, miR-511-3p, Let-7a-5p, miR-16-5p, miR-146a-5p, miR-142-3p. Furthermore, in order to demonstrate that miR-375 is not endogenously expressed by macrophages but transferred from the cancer cells, the authors use actinomycin D to block miR-375 transcription. As proof of their method the authors show that actinomycin D can efficiently block PPAR-gamma transcription. However, it is not shown that actinomycin D can effectively block miR-375 expression in macrophages. Therefore, it would be more reliable to use miR-375 KO macrophages or Dicer-deficient macrophages as used in other studies (Squadrito et al., Cell Reports 2014) to formally show that miR-375 measured upon exposure to cancer cell medium is not endogenously expressed by macrophages.
2. The significance of miR-375 transfer for TAM recruitment/abundance in human cancer is not explored. The authors should examine whether miR-374 expression correlates with TAM-specific markers or cytokines that promote TAM infiltration in human tumors, such as CCL2, CSF1 and GM-CSF?
3. The mechanism by which miR-375 impairs TAM infiltration to tumors is not fully addressed in this manuscript. For example, it is not clear if miR-375 is transferred only to TAMs or also to monocytes, including monocytes in the spleen or in the bone marrow. Of note transfer of miR-375 to TAMs will not impact on their tumor infiltration since these cells are already in the tumors. Furthermore, is miR-375 impairing TAM infiltration by conditioning CCL2 production in the cancer cells or by downregulating PXN and TNS3 expression in macrophages?
4. The authors suggest that miR-375 is not loaded into extracellular vesicles as RNase A treatment results in its degradation. However, the particles containing miR-375 were not characterized in this study. The authors should characterize better these particles, including size and composition.
5. The control groups used by the authors in many of the experiments are not indicated. The authors should be aware that for experiments using decoy sequences the most indicated control should be a scrambled decoy sequence. The authors should describe properly the control groups that they use in their experiments and possibly change the term 'control' for the exact condition that is used in that experiment.

GENERAL RESPONSE:

We are grateful for the constructive and encouraging comments of the reviewers and their stimulating experimental advice to improve our manuscript. Addressing their queries certainly improved our manuscript. We apologize for the delay in resubmitting the revised version, which took longer than expected because of time-demanding new experiments and establishment of new technology in our lab. Please find our point-by-point response below.

Reviewer #1, Expertise: apoptosis, TME (Remarks to the Author):

This is a rigorous and robust study which demonstrates the importance of miR-375 in altering the phenotype of TAMs in breast cancer and suggesting that transfer of miR-375, especially from apoptotic tumor cells to monocytes and macrophages can profoundly influence their migratory and activation states to promote tumor growth. The authors dissect underlying mechanisms through in vitro and in vivo approaches and demonstrate (1) a molecular mechanism (macrophage CD36) for uptake of miR-375 which they prove to be non-vesicular, and (2) miR-375 target genes (macrophage PXN and TNS3). Of course this type of discovery research generates many additional questions - such as those raised by the authors in their Discussion - but the level of detailed new knowledge elucidated in this paper warrants publication after attention to minor issues, in this referee's opinion.

Minor points:

1. MCF-7 cells, on which the authors' conclusions are largely based, are unusual in that they are deficient in the effector caspase-3 which may result in subtly different apoptotic features than caspase-3-expressing tumor cells. Were the same observations on production and transfer of miR-375 seen with ACM from caspase-3+ tumor cells?

RESPONSE: *Yes, we also noticed the release and transfer of miR-375 by caspase-3 positive MDA-MB-231 and T-47D cells. Information on the expression of miR-375 in caspase-3 negative MCF-7 cells as well as caspase-3 positive T-47D and MDA-MB-231 cells (Supplementary Fig. 3a) and its transfer to macrophages during coculture (Fig. 1k) is now included. Specifically, we compared miR-375 release from viable and apoptotic MCF-7, T-47D and MDA-MB-231 cells with the notion that caspase-3 deficiency neither influenced miR-375 production (Supplementary Fig. 3a) nor its release upon staurosporine-mediated apoptosis (Fig. 2d) or transfer to macrophages during coculture (Fig 1k). This data now is mentioned in Results.*

2. Is CCL2 production increased in macrophages as a consequence of miR-375 activity?

RESPONSE: *We observed a positive correlation between miR-375 levels and CCL2 production in MCF-7 cells and TAMs (Fig. 7d & Supplementary Fig. 9c). As suggested by*

this reviewer we validated this relationship and scanned various breast cancer cell lines including human mammary epithelial cells (HMEC) for their miR-375 levels compared to CCL2 mRNA and protein expression. Our initial observation is now consolidated with new data in Supplementary Fig. 3a and Supplementary Fig. 5a, suggesting that miR-375 levels are directly proportional to CCL2 mRNA expression in breast cancer cells and HMEC. We also analyzed miR-375 release in ACM and VCM of various breast cancer cell lines including HMEC, compared to CCL2 release in these samples. There is a positive correlation between miR-375 and CCL2 release in VCM and ACM of these cells (Fig. 2d and Supplementary Fig. 5b). Macrophages sorted from 3D tumor spheroids of decoy MCF-7 cells show a lower CCL2 expression compared to macrophages from control spheroids (Supplementary Fig. 9c). To more directly address the question of this reviewer, we also transfected primary human macrophages with miR-375 mimic or scramble control and treated them with ACM of MCF-7 cells. As presented in Supplementary Fig. 5e, miR-375 levels in macrophages were directly proportional to CCL2 mRNA expression. These evidences suggest a direct proportional relationship of miR-375 levels with CCL2 expression and release in both, tumor cells and macrophages.

Reviewer #2, Expertise: miRNA, cancer (Remarks to the Author):

In the manuscript by Frank et al. the authors report that apoptotic breast cancer cells, independent of exosomes, release miR-375, which is taken up by tumor-associated macrophages (TAM) via the CD36 scavenger receptor. In TAM, miR-375 is accumulated and regulates TNS3 and PXN gene expression to promote infiltration into the tumor spheroid. In tumor cells, miR-375 regulates CCL2 through an indirect mechanism to recruit macrophages. The work is original and provides new understandings on the regulation of macrophage infiltration and the interaction between tumor and immune cells via extracellular RNA. The findings are novel and experiments are well designed. Data are of high quality and support the conclusion. Statistic methods are appropriate. However the following concerns need to be addressed.

1. It was previously reported that miR-375 expression level is associated with ER status in breast cancer (Cancer Res. 2010;70:9175-84). In addition to T47D, MCF-7, and MDA-MB-231, more breast cancer cell lines (including both ER+ and ER-) and primary breast epithelial cells should be examined for their function in transferring miR-375 and recruiting macrophages in order to determine if this effect is specific to ER+ tumors.

RESPONSE: *As suggested, we now included more ER+ and ER- breast cancer cell lines along with primary breast epithelial cells and measured miR-375 expression (new*

Supplementary Fig. 3a). We also used candidate ER+ (MCF-7, T-47D, EFM-192A) and ER- (MDA-MB-468, MDA-MB-231, SKBR3, HCC1937) breast cancer cell lines and primary human breast epithelial cells for coculture experiments with primary human macrophages. Specifically, we used these cells for ACM production, to subsequently stimulate human macrophages. ACM of ER+ breast cancer cell lines showed a higher release of miR-375 compared to ER- cells (Fig. 2d), which also reflects their miR-375 expression levels (Supplementary Fig. 3a). Along these lines, miR-375 expression in ER+ cell lines was higher compared to ER- cell lines. Data in the new Fig. 1k shows that ER expression in breast cancer cell lines is proportional to their miR-375 transfer potency to primary human macrophages during coculture. We also generated 3D tumor spheroids of ER+ (MCF-7, T47D) and ER- (MDA-MB-468, MDA-MB-231) cells (New Supplementary Fig. 10a) and accessed infiltration of human PBMC CD14⁺ cells (New Supplementary Fig. 10c). We measured reduced infiltration of CD14⁺ cells in ER- tumor spheroids as compared to ER+ spheroids (new Supplementary Fig. 10b), which may point towards a secondary role of ER expression in macrophage infiltration in 3D spheroids.

2. Primary breast tumors should be examined for a correlation between miR-375 level and macrophage markers. This should also be compared between ER+ and ER- tumor.

RESPONSE: In general, there are very limited datasets exploring mRNA and miR expression of the same tumor and adjacent normal tissue. Moreover, these datasets comprise gene expression of whole tumor, with a limited value in accessing cell specific gene expression. To circumvent this problem, we adopted a challenging approach and measured cell specific miR-375 expression on tissue microarray (TMA) slides of breast cancer patients (CHTN, University of Virginia) by in situ hybridization using double DIG labeled miRCURY LNATM miRNA detection probes. It was a major task to combine miR-375 in situ hybridization with immunofluorescent staining of macrophage markers. We succeeded to stain TMA slides (n = 155 patients; n = 49 normal breast) with the tissue macrophage marker MERTK^{1,2} using a multispectral imaging system (PhenOptics, Perkin Elmer) that allows analyzing the expression of MERTK via tyramide signal amplification, automated slide processing and analysis. As presented in Fig. 9c, there is a positive correlation of miR-375 levels and MERTK expression in tumor sections with Pearson's $r = 0.67$ and $p < 0.001$. We believe that tissue microarrays of mammary carcinoma patients substantiated our conclusion that enhanced miR-375 levels increase monocyte/macrophage migration in the tumor microenvironment and adds clinical relevance to our findings.

Correlating miR-375 expression with the ER status of breast tumor has already been reported in a few studies³⁻⁵. In breast cancer cell lines there was an enhanced expression of miR-375 in ER+ cells, which we substantiated in our study (Supplementary Fig. 3a, Fig. 2d).

However, the interesting study by de Souza Rocha Simonini et al., also referenced by this reviewer, presented data from only 4 ER+ and 4 ER- breast cancer patients, showing no correlation between miR-375 expression and ER status³. A similar picture emerged when analyzing public available TCGA datasets on invasive breast carcinoma, which were obtained through cbiportal.org (See Figure R1 below). Data from two studies dealing with invasive breast carcinoma and involving 766⁶ and 777⁷ patients, respectively show no correlation between miR-375 expression and the ER (Fig. R1a, c) or HER2 status (Fig. R1b, d).

Fig. R1 Correlation of miR-375 expression and the ER or HER2 status. TCGA datasets were acquired via cbiportal.org and analyzed for miR-375 expression and **a,c** estrogen receptor (ER) and **b,d** human epidermal growth factor receptor 2 (HER2) status.

Our *in situ* hybridization data on miR-375 expression in invasive breast carcinoma tissue microarray slides show a significantly enhanced expression of miR-375 in invasive breast carcinoma and DCIS tissues compared to normal breast tissue (Fig. 9a, b, d). Regarding the ER status, our data confirm the above-mentioned studies and datasets, not showing any correlation between miR-375 expression and the ER status (Fig. 9e). However, there was a significantly enhanced level of miR-375 in HER2- samples compared to HER2+ (Fig. 9f).

Our study suggests that the role of miR-375 would be more relevant with respect to monocyte/macrophage infiltration at early stages of tumorigenesis, which is difficult to capture in clinical samples. Datasets that comprises late stage samples of tumorigenesis may not reflect the full potential of miR-375 in breast carcinogenesis. This argument is supported by Tsai et al⁴, showing that miR-375 was upregulated in very young ER+ patients (<35 years) and was one of the candidates miR in the age-onset group.

3. The protein levels of PXN and TNS3 are never shown in the manuscript and should be added.

***RESPONSE:** Protein levels of PXN and TNS3 are now measured in primary human macrophages after 48h treatment with ACM derived from apoptotic MCF-7 cells (Fig. 5b, Supplementary Fig. 7d). Corroborating mRNA data, we saw attenuated protein expression of PXN and TNS3 upon overexpression of miR-375 in primary human macrophages (Fig. 5a, Supplementary Fig. 7d). Furthermore, we show that PXN and TNS3 protein expression can be rescued by transfecting target site blockers that compete and block the binding of miR-375 to the 3' UTRs of PXN and TNS3 (Fig. 6b-e, Supplementary Fig. 8a, b).*

4. To confirm PXN and TNS3 indeed mediate miR-375's effect the authors should use PXN and TNS3 expression vectors with mutated miR-375 binding sites for restoration assay.

***RESPONSE:** We adopted a method that maintained physiological level of PXN and TNS3 in macrophages and allowed to interrogate the role of these proteins in context of miR-375 during macrophage migration. A direct causal relationship of miR-375, PXN and TNS3 in macrophage migration was demonstrated using miR-375 target site blockers (TSB)⁸, which essentially negate the effects of miR-375 overexpression, either by mimic or ACM treatment, on downregulation of PXN and TNS3. Custom-designed TSBs with phosphorothioate backbone modifications from Exiqon (miRCURY LNATM microRNA TSB) were used (Fig. 6). TSB sequences are designed with a large arm that covers the miR binding site and a short arm outside the miR seed to ensure target specificity to 3'UTR of PXN and TNS3 (Fig. 6a). We rescued mRNA and protein downregulation of PXN and TNS3 elicited by miR-375 mimic or ACM treatment by using TSBs (Fig. 6b-e). Restoring expression of these proteins had decisive effects on macrophage migration in response to ACM in scratch assays (Fig. 6f, g). This miR-375 loss-of-function assay and rescue of PXN and TNS3 expression clearly established their critical involvement in macrophage migration.*

5. The mechanism through which miR-375 induces CCL2 is not identified. Since ER+ breast cancer cells express more miR-375, it would be important to test if they also produce more CCL2.

***RESPONSE:** We scanned various ER+ and ER- breast cancer cell lines including primary human mammary epithelial cells (HMEC) for their miR-375 levels in relation to CCL2 mRNA and protein expression. We consolidated our data (new Supplementary Fig. 3a and Supplementary Fig. 5a), suggesting that miR-375 levels in ER+ (MCF-7, T-47D, EFM-192A) and ER- (MDA-MB-468, MDA-MB-231, SKBR3, HCC1937) breast cancer cell lines as well as HMEC were directly proportional to CCL2 mRNA expression in these cells. We also*

measured miR-375 release in ACM and VCM of various ER+ and ER- breast cancer cell lines compared to the CCL2 release in these samples. Again, the miR-375 release from VCM and ACM is directly proportional to the CCL2 release in these samples (Fig. 2d and Supplementary Fig. 5b). ER+ breast cancer cells such as EFM-192A, which show highest miR-375 levels (Supplementary Fig. 3a) also show highest CCL2 expression/release during staurosporine-mediated apoptosis (Supplementary Fig. 5a, b).

6. miR-183-5p also seems to be transferred from breast cancer cells to macrophages. What's its function? At the very least, does it synergize with miR-375 in stimulating and recruiting macrophages?

RESPONSE: Indeed, we see miR-183-5p transfer from tumor cells to macrophages. The transfer mechanism most likely is via uptake of exosomes as this miR was protected during RNase treatment of ACM (Supplementary Fig. 4a). Based on the *in silico* targetome analysis, we hypothesized that miR-183-5p may not participate in macrophage migration. We used mimics to overexpress miR-183-5p in primary human macrophages alone or in combination with miR-375 (Fig. R2a) and subjected control- or mimic-transfected macrophages to scratch assays (Fig. R2b). As documented in Fig. 4, mere overexpression of miR-375 without any secondary stimulus does not influence migration (Fig. R2c). However, overexpression of this miR along with stimulation of MCF-7 ACM enhances macrophage migration. On the other hand, overexpression of miR-183-5p alone or in combination with miR-375 does not increase migration of ACM-stimulated macrophages (Fig. R2b, c). Furthermore, unlike miR-375, which downregulated PXN and TNS3 mRNA expression, miR-183-5p left their expression unaltered (Fig. R2d). Apparently, miR-183-5p neither synergizes nor antagonizes miR-375-mediated macrophage migration *in vitro*.

Fig. R2. Role of miR-183-5p in macrophage migration. (a - d) Primary human M Φ were transfected with synthetic miR-375/miR-183-5p mimic or negative cel-miR-39a control for 24 h, followed by treatment with MCF-7 cell ACM for 30 min. Cells were washed and fresh M Φ media was added for 24 h. **a** Relative miR-375 and miR-183-5p abundance. **b** Scratches were generated with a small pipette tip in a marked area. Pictures were taken at 0 and 24 h and the cell free area within the scratch was measured using ImageJ software. **c** Percentage gap closure after 24 h was calculated with respect to gap area at 0 h and normalized to untreated M Φ control (M Φ control = 0%). Data are means \pm SEM of $n = 8$. P -values were calculated using two-way ANOVA with Bonferroni's correction. *, $p < 0.05$. **d** PXN and TNS3 mRNA expressions in M Φ were measured by qPCR and normalized to scramble transfected M Φ . Data are means \pm SEM of $n = 4$. P -values of **d** were calculated using one-sample t test *, $p < 0.05$, **, $p < 0.01$, ***, $p < 0.001$.

Reviewer #3, Expertise: TAMs (Remarks to the Author):

In this manuscript, Frank and coworkers show that breast cancer cells transfer miR-375 to macrophages to promote their infiltration to tumors. The authors show that miR-375 is

transferred to macrophages either alone or in complex with LDL, but not in association with extracellular vesicles. According to the authors, macrophages internalize miR-375 through CD36. The authors then show that cancer cell-derived miR-375 downregulates the expression of PXN and TNS3 in macrophages. PXN and TNS3 are two novel target genes of miR-375. Furthermore, miR-375 enhances CCL2 production in cancer cells. In a tumor mouse model of breast cancer, knockdown of miR-375 in cancer cells using a decoy strategy caused Pxn and Tns3 upregulation in tumor-associated macrophages (TAMs) while decreasing macrophage and monocyte infiltration.

The manuscript is well written and the findings are novel and interesting. However, there are some aspects that are still at an early stage and should be further developed by the authors before this manuscript is considered for publication.

Comments:

1. The authors claim that cancer cell-derived miR-375 modulates gene expression in macrophages upon its internalization. Previous studies have demonstrated that only miRNAs expressed at a relevant level can modulate gene expression (Brown et al., Nature Biotechnology 2007). In order to demonstrate that exogenous miR-375 can reach functional levels in macrophages, the authors should quantify the absolute expression level (number of copies) of miR-375 compared to other miRNAs, which are relevant to macrophage biology, such as miR-21-5p, miR-511-3p, Let-7a-5p, miR-16-5p, miR-146a-5p, miR-142-3p.

RESPONSE: We measured copy numbers of candidate miRs (let-7a-5p, miR-142-3p, miR-33a, miR-511-3p, miR-21-5p and miR-375) in control and ACM (from MCF-7 cells)-treated human macrophages (see Figure below, Fig. R3). miR-375 content in ACM-treated macrophages was comparable with miR-511-3p, whereas expression was several fold lower compared to let-7a-5p, miR-142-3p and miR-21-5p. Nevertheless, miR-375 content in ACM-treated primary human macrophages was physiologically relevant and functional⁹, as it was sufficient to repress PXN and TNSs, both at mRNA and protein level (Fig. 5).

Fig. R3. miR copy numbers in human primary macrophages. MΦ were treated with ACM of MCF-7 cells for 30 min (ACM) or left untreated (control). Cells were washed, fresh media was added for 24 h, cells were harvested and the number of copies per ng total RNA was calculated for individual miRs. Data are means ± SEM of n = 6.

Furthermore, in order to demonstrate that miR-375 is not endogenously expressed by macrophages but transferred from the cancer cells, the authors use actinomycin D to block miR-375 transcription. As proof of their method the authors show that actinomycin D can efficiently block PPAR-gamma transcription. However, it is not shown that actinomycin D can effectively block miR-375 expression in macrophages. Therefore, it would be more reliable to use miR-375 KO macrophages or Dicer-deficient macrophages as used in other studies (Squadrito et al., Cell Reports 2014) to formally show that miR-375 measured upon exposure to cancer cell medium is not endogenously expressed by macrophages.

RESPONSE: We paid full attention to these comments and addressed them in the following way: Macrophages were pre-treated with 2.5 µg/ml of actinomycin D for 3h, which essentially blocks all transcriptional activity in macrophages, prior to subjecting them to a coculture with tumor cells. Actinomycin D efficacy was confirmed by the decreased expression of the short-lived PPARG mRNA (Fig. 1f). In contrast, actinomycin D failed to block elevation of miR-375 levels in cocultured macrophages (Fig. 1e). This, in addition to the very low basal miR-375 expression in macrophages and lack of induction with various other stimuli (Fig. 1d), argues against de novo induction of miR-375 in macrophages. As suggested by the reviewer, we knocked down (KD) DICER in primary human macrophages using siRNA (Fig. 1g) and then performed cocultures with tumor cells. DICER KD significantly reduced miR-21-5p and miR-142-3p expression in macrophages (Fig. 1h), however it did not affect miR-375 elevation in macrophage after their coculture with MCF-7 cells (Fig. 1i). Furthermore, pre-mir-375 expression also remained unaltered in macrophages during the coculture with tumor cells and DICER KD (Fig. 1j). These data, in addition to Fig. 2a-c, strongly suggest that miR-375 is not induced in macrophages, rather is tumor cell-derived.

2. The significance of miR-375 transfer for TAM recruitment/abundance in human cancer is not explored. The authors should examine whether miR-374 expression correlates with TAM-specific markers or cytokines that promote TAM infiltration in human tumors, such as CCL2, CSF1 and GM-CSF?

RESPONSE: As mentioned in response to Reviewer #2, we performed in situ hybridization to detect miR-375 in tissue microarray slides of invasive breast carcinoma patients. Since CCL2 is a secretory protein, immunohistochemical/immunofluorescent detection of intracellular CCL2 not may provide the entire picture. Furthermore, IHC/IF cannot be used to colocalize CCL2 with small non-coding RNAs such as miR-375. Therefore, we performed a time-consuming task to standardized RNAscope® staining of CCL2 and miR-375 in situ hybridization. Data presented in Fig. 9g show a positive correlation between miR-375 and CCL2 expression in invasive mammary carcinoma. We also established multispectral

imaging in our lab (PhenOptics, Perkin Elmer) to analyze expression of the macrophage marker MERTK^{1,2} via tyramide signal amplification, automated slide processing and analysis ($n = 155$ patients; $n = 49$ normal breast). As seen in Fig. 9c, there is a positive correlation of miR-375 levels and MERTK expression in tumor sections with Pearson's $r = 0.67$ and $p < 0.001$. We observed colocalization of CCL2 and miR-375 in MERTK⁺ tumor-associated macrophages (Fig. 9h).

Our study highlights a role of miR-375 in early stages of mammary carcinogenesis, which may not be recapitulated in grade II, III tissue samples. However, a positive correlation of miR-375 with the tissue specific macrophage marker MERTK and monocyte/macrophage chemoattractant CCL2 in invasive mammary carcinoma patients substantiated our findings and adds some clinical relevance to our findings.

3. The mechanism by which miR-375 impairs TAM infiltration to tumors is not fully addressed in this manuscript. For example, it is not clear if miR-375 is transferred only to TAMs or also to monocytes, including monocytes in the spleen or in the bone marrow. Of note transfer of miR-375 to TAMs will not impact on their tumor infiltration since these cells are already in the tumors. Furthermore, is miR-375 impairing TAM infiltration by conditioning CCL2 production in the cancer cells or by downregulating PXN and TNS3 expression in macrophages?

RESPONSE: We agree that it is important to understand in which compartment miR-375 is taken up and whether monocytes do so as well. To address this we used, compared to MCF-7 cells, a more aggressive mouse mammary carcinoma cell line E0771 with a stable KD of miR-375 using a lentiviral decoy construct or control empty vector (Fig. 8g). These cells were orthotopically transplanted into mammary fat pad gland 3 and 8 of immunocompetent BL/6 mice. Palpable tumors were harvested after 2 weeks along with spleen, bone marrow and blood (Fig. 8h). Monocytes and macrophages were FACS sorted from these samples (Supplementary Fig. 11c) and miR-375 content was measured by real-time qPCR (Fig. 8k). RNA isolated from plasma of these mice served as a source of miR-375. In agreement with the nude mice model and human MCF-7 breast cancer cells (Fig. 8a), we noticed reduced monocyte and macrophage infiltration in decoy tumors compared to control tumors (Fig. 8i). miR-375 levels in the plasma of mice receiving decoy E0771 cells were significantly lower compared to mice receiving control E0771 cells (Fig. 8j). As expected, there were reduced levels of miR-375 in blood monocytes, as well as in monocytes and macrophages from decoy tumors compared to control tumors. Interestingly, we detected miR-375 in bone marrow monocytes and macrophages, with levels reflecting the situation in other organs such as spleen, where reduced miR-375 levels were observed in cells of decoy tumors (Fig. 8k). Data from this mouse model suggest that the systemic

release and uptake of miR-375 by monocytes and macrophages may have far reaching consequences in addition to other tumor-derived factors.

Concerning whether miR-375 increases TAM infiltration by coordinating CCL2 production in cancer cells or by downregulating PXN and TNS3 in macrophages, our data suggests that both events are coupled. We separated these events in an *in vitro* setting, when we restored PXN and TNS3 expression by specific miR-375 target site blockers in macrophages, despite high miR-375 levels (Fig. 6b-e). In this setting, despite stimulating with CCL2-containing ACM (Supplementary Fig. 5b) macrophages failed to migrate (Fig. 6f, g). However, monocyte migration in a transwell chemotaxis assay clearly demanded CCL2 in ACM (Fig. 4a, b). Thus, our data suggest that both, PXN/TNS3 and CCL2 expression are important for macrophage migration and underscore the pleiotropic effects of miR-375 in this process.

4. The authors suggest that miR-375 is not loaded into extracellular vesicles as RNAse A treatment results in its degradation. However, the particles containing miR-375 were not characterized in this study. The authors should characterize better these particles, including size and composition.

RESPONSE: We reached out to characterize miR-375 containing particles. Considering published information that miR-375 is associated with LDL in the serum of hypercholesterolemia¹⁰, we prepared ACM of MCF-7 cells with fetal calf serum as a source of LDL or in serum-free media. As expected, miR-375 levels were significantly higher in serum (as the potential source of HDL and LDL) containing ACM compared to serum-free ACM. In contrast, the level of miR-183-5p, which localizes to exosomes, remained unaffected by the presence or absence of serum (Fig. 3a). This pointed towards critical serum factors in stabilizing and transporting miR-375. Therefore, we isolated HDL and LDL fractions from these ACM samples using the method described by Yamamoto *et al.*¹¹ and used them as a source of RNA for qPCR detection of miR-375. miR-375 was exclusively present in the LDL fraction of ACM. Both, LDL and HDL fractions were devoid of control candidate miR-183-5p (Fig. 3b), which is vesicular bound and RNAase protected (Supplementary Fig. 4a, b). These experiments suggest that miR-375 is LDL bound, which underscores the role of CD36 for miR-375 uptake.

5. The control groups used by the authors in many of the experiments are not indicated. The authors should be aware that for experiments using decoy sequences the most indicated control should be a scrambled decoy sequence. The authors should describe properly the control groups that they use in their experiments and possibly change the term 'control' for the exact condition that is used in that experiment.

RESPONSE: As suggested by the reviewer, controls are more clearly explained in figure legends and the Method section.

References:

1. Ringleb, J. *et al.* Apoptotic Cancer Cells Suppress 5-Lipoxygenase in Tumor-Associated Macrophages. *Journal of immunology (Baltimore, Md. : 1950)* **200**, 857–868; 10.4049/jimmunol.1700609 (2018).
2. Zizzo, G., Hilliard, B. A., Monestier, M. & Cohen, P. L. Efficient clearance of early apoptotic cells by human macrophages requires “M2c” polarization and MerTK induction. *Journal of immunology (Baltimore, Md. : 1950)* **189**, 3508–3520; 10.4049/jimmunol.1200662 (2012).
3. Souza Rocha Simonini, P. de *et al.* Epigenetically deregulated microRNA-375 is involved in a positive feedback loop with estrogen receptor alpha in breast cancer cells. *Cancer research* **70**, 9175–9184; 10.1158/0008-5472.CAN-10-1318 (2010).
4. Tsai, H.-P., Huang, S.-F., Li, C.-F., Chien, H.-T. & Chen, S.-C. Differential microRNA expression in breast cancer with different onset age. *PLoS one* **13**, e0191195; 10.1371/journal.pone.0191195 (2018).
5. Zhou, X. *et al.* MicroRNA-9 as potential biomarker for breast cancer local recurrence and tumor estrogen receptor status. *PLoS one* **7**, e39011; 10.1371/journal.pone.0039011 (2012).
6. Comprehensive molecular portraits of human breast tumours. *Nature* **490**, 61–70; 10.1038/nature11412 (2012).
7. Ciriello, G. *et al.* Comprehensive Molecular Portraits of Invasive Lobular Breast Cancer. *Cell* **163**, 506–519; 10.1016/j.cell.2015.09.033 (2015).
8. Sonnevile, F. *et al.* MicroRNA-9 downregulates the ANO1 chloride channel and contributes to cystic fibrosis lung pathology. *Nature communications* **8**, 710; 10.1038/s41467-017-00813-z (2017).
9. Ouimet, M. *et al.* MicroRNA-33-dependent regulation of macrophage metabolism directs immune cell polarization in atherosclerosis. *The Journal of clinical investigation* **125**, 4334–4348; 10.1172/JCI81676 (2015).
10. Vickers, K. C., Palmisano, B. T., Shoucri, B. M., Shamburek, R. D. & Remaley, A. T. MicroRNAs are transported in plasma and delivered to recipient cells by high-density lipoproteins. *Nature cell biology* **13**, 423–433; 10.1038/ncb2210 (2011).
11. Yamamoto, H. *et al.* VLDL/LDL acts as a drug carrier and regulates the transport and metabolism of drugs in the body. *Scientific Reports* **7**, 633; 10.1038/s41598-017-00685-9 (2017).

REVIEWERS' COMMENTS:

Reviewer #1 (Remarks to the Author):

The authors have greatly improved their manuscript by adequately addressing the points raised.

Reviewer #2 (Remarks to the Author):

The authors have successfully addressed all my previous concerns. This is a well designed and presented study.

Reviewer #3 (Remarks to the Author):

Frank and coworkers have now addressed all my concerns thus improving the overall quality of the manuscript. Nevertheless, I suggest the authors adding Figure R 3 to the article, in order to indicate the contribution of exogenous miR-375 to the pool of endogenous miRNAs in macrophages. The revised manuscript is now suitable for publication.

Reviewer #1 (Remarks to the Author):

The authors have greatly improved their manuscript by adequately addressing the points raised.

RESPONSE: *Thank you!*

Reviewer #2 (Remarks to the Author):

The authors have successfully addressed all my previous concerns. This is a well designed and presented study.

RESPONSE: *Thank you!*

Reviewer #3 (Remarks to the Author):

Frank and coworkers have now addressed all my concerns thus improving the overall quality of the manuscript. Nevertheless, I suggest the authors adding Figure R 3 to the article, in order to indicate the contribution of exogenous miR-375 to the pool of endogenous miRNAs in macrophages. The revised manuscript is now suitable for publication.

RESPONSE: *As suggested, Figure R3 is now moved to the article as Supplementary Fig. 7c and is also mentioned in Results section.*